# A theory of joint attractor dynamics in the hippocampus and the entorhinal cortex accounts for artificial remapping and grid cell field-to-field variability

**Haggai Agmon[1], Yoram Burak[1,2]***

[1]Edmond and Lily Safra Center for Brain Sciences, The Hebrew University of Jerusalem, Jerusalem, Israel; [2]Racah Institute of Physics, The Hebrew University of Jerusalem, Jerusalem, Israel

**Abstract** The representation of position in the mammalian brain is distributed across multiple neural populations. Grid cell modules in the medial entorhinal cortex (MEC) express activity patterns that span a low-dimensional manifold which remains stable across different environments. In contrast, the activity patterns of hippocampal place cells span distinct low-dimensional manifolds in different environments. It is unknown how these multiple representations of position are coordinated. Here, we develop a theory of joint attractor dynamics in the hippocampus and the MEC. We show that the system exhibits a coordinated, joint representation of position across multiple environments, consistent with global remapping in place cells and grid cells. In addition, our model accounts for recent experimental observations that lack a mechanistic explanation: variability in the firing rate of single grid cells across firing fields, and artificial remapping of place cells under depolarization, but not under hyperpolarization, of layer II stellate cells of the MEC.

***For correspondence:**
yoram.burak@elsc.huji.ac.il

**Competing interests:** The authors declare that no competing interests exist.

## Introduction

The discoveries of spatially selective cells in the hippocampus (*O'Keefe and Dostrovsky, 1971*) and the medial entorhinal cortex (MEC) (*Hafting et al., 2005*), have opened a window into the mechanisms underlying neural coding and processing of a high-level cognitive variable, which is separated from direct sensory inputs: the brain's estimate of an animal's position. Experimental evidence and theoretical reasoning have suggested that the neural representation of this variable is maintained by multiple attractor networks in the hippocampus (*Battaglia and Treves, 1998*; *Quirk et al., 1990*; *Samsonovich and McNaughton, 1997*; *Schlesiger et al., 2018*; *Wills et al., 2005*) and the MEC (*Burak, 2014*; *Burak and Fiete, 2009*; *Fuhs and Touretzky, 2006*; *Guanella et al., 2007*; *McNaughton et al., 2006*; *Stensola et al., 2012*), each expressing highly restricted patterns of neural activity at the population level that are preserved across multiple environments and conditions (*Gardner et al., 2019*; *Trettel et al., 2019*; *Yoon et al., 2013*). Under this interpretation of the experimental observations, patterns of population activity in the hippocampal-entorhinal system are selected from a limited repertoire, shaped by synaptic connectivity that enforces the attractor dynamics. These activity patterns are dynamically linked with the animal's position in response to external sensory inputs and self-motion cues.

Despite the hypothesized role of recurrent connectivity in shaping spatial responses in the hippocampus and the MEC, theoretical research on the relationship between these brain areas has mostly treated them as successive stages in a processing hierarchy involving feed-forward connectivity from grid cells to place cells, or vice-versa (*Figure 1a*) (but see [*Laptev and Burgess, 2019*; *Rennó-Costa and Tort, 2017*]). It is easy to show that summed inputs from multiple grid cells, combined

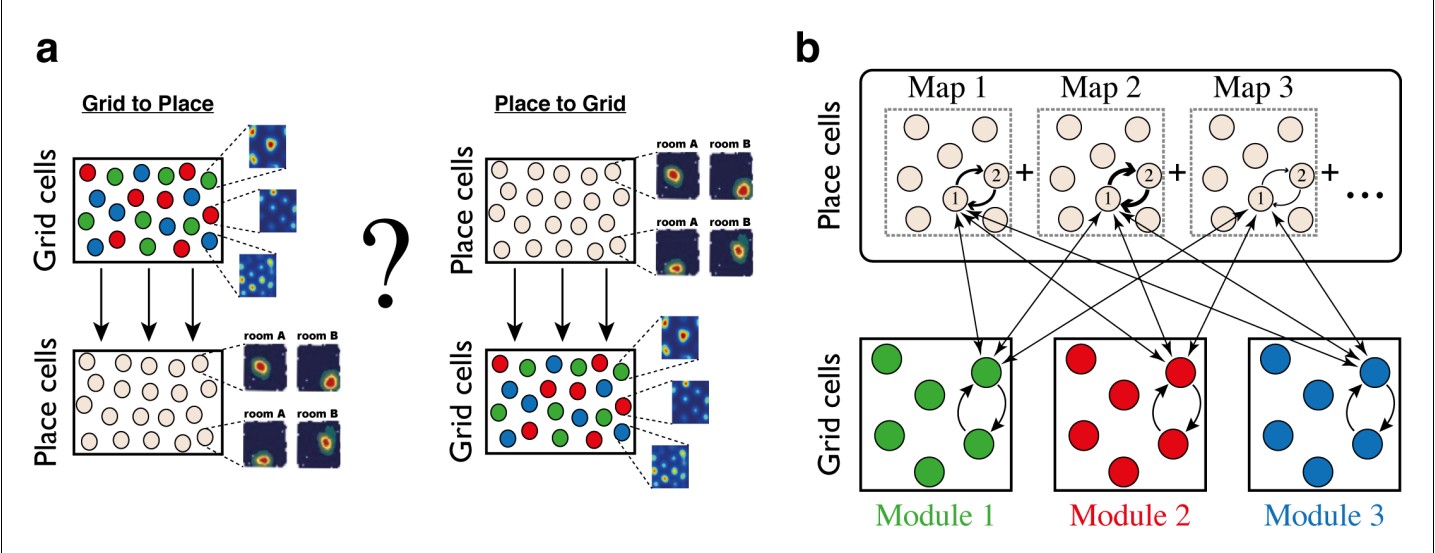

**Figure 1.** Schematic illustration: possible architectures of grid cell and place cell connectivity. (**a**) Two hypotheses on connectivity, with hierarchical relationship between grid cells and place cells. Left: feed-forward connectivity from grid cells to place cells. Right: feed-forward connectivity from place cells to grid cells. Grid cells are color coded by their allocation to modules. Place cells exhibit global remapping between distinct environments (room A/room B) accompanied by grid cells realignment (not shown). (**b**) Schematic illustration of the model's architecture: place cells and grid cells are bidirectionally coupled. Top: the strength of synaptic connectivity between a pair of place cells is a sum of map-specific contributions arising from the distinct maps. In each discrete map, the connectivity depends on the environment-specific spatial separation between the place cell receptive fields (arrow thicknesses for place cells #1 and #2). Bottom: grid cell modules are modeled as independent continuous attractors. Recurrent connectivity is identical in all modules. Straight arrows: the bi-directional connection between a grid cell and a place cell is proportional to the overlap between their receptive fields across the discrete maps.

with thresholding or competitive dynamics, can produce place-cell like responses (*de Almeida et al., 2009*; *Monaco et al., 2011*; *Neher et al., 2017*; *Rolls et al., 2006*; *Solstad et al., 2006*). It is also straightforward to produce grid-cell like responses by combining synaptic inputs from multiple place cells whose receptive fields are periodically arranged in space. Furthermore, theoretical studies have demonstrated that plasticity mechanisms, acting between place cells and their post-synaptic partners in the MEC, can give rise to synaptic connectivity of this form, thereby producing grid-cell like responses that are driven by place cell inputs (*D'Albis and Kempter, 2017*; *Dordek et al., 2016*; *Kropff and Treves, 2008*; *Monsalve-Mercado and Leibold, 2017*; *Stepanyuk, 2015*; *Weber and Sprekeler, 2018*). Nevertheless, the functional relationship between grid cells and place cells, its relationship with recurrent connectivity within each brain region, and its role in the encoding of positions across multiple environments, remain unclear.

There are several difficulties with the hypothesis that spatial selectivity in the entorhinal-hippocampal system emerges from a simple feedforward architecture. First, a number of experiments indicate that grid cell inputs are not necessary for emergence of place cell spatial specificity. Stable place fields are expressed in the hippocampus of preweanling rats before the appearance of stable grid cell activity in the MEC (*Langston et al., 2010*; *Wills et al., 2010*), and place fields are maintained under MEC lesions (*Hales et al., 2014*). Global hippocampal remapping can occur without input from the MEC (*Schlesiger et al., 2018*), and place fields are expressed under the inactivation of the medial septum, which eliminates the spatial periodicity of grid cells (*Koenig et al., 2011*). Furthermore, new place fields can be established in novel environments under medial septum inactivation (*Brandon et al., 2014*).

Second, there are difficulties with the opposite hypothesis, that feedforward inputs from place cells are the sole determinants of grid cell specificity. One such difficulty arises from the observation that place cell activity patterns remap in different environments (*Muller and Kubie, 1987*). If grid cell responses emerge under activity dependent plasticity of feedforward synapses from place cells to grid cells (*D'Albis and Kempter, 2017*; *Dordek et al., 2016*; *Kropff and Treves, 2008*; *Monsalve-Mercado and Leibold, 2017*; *Stepanyuk, 2015*; *Weber and Sprekeler, 2018*), it is difficult to

explain why they remain grid-like across familiar and novel environments, and why phase relationships between grid cells within a module are tightly preserved (*Gardner et al., 2019*; *Trettel et al., 2019*; *Yoon et al., 2013*). Furthermore, grid cells maintain their correlation structure under hippocampal inactivation (*Almog et al., 2019*), indicating that the low dimensionality of activity within each grid cell module is independent of hippocampal inputs.

Third, anatomically, the MEC and the hippocampus are reciprocally connected (*van Strien et al., 2009*). A significant proportion of excitatory projections to place cells that arise in MEC originate in grid cells (*Zhang et al., 2013*), and there are abundant feedback projections from place cells to deep layers of MEC (*Deadwyler et al., 1975*).

Here, instead of assuming a feed-forward architecture of the entorhinal-hippocampal network, we explore theoretically an alternative hypothesis, that grid cells and place cells are driven by two types of synaptic inputs (*Figure 1b*): inputs arising from recurrent connectivity within each brain area that restrict activity to lie within low dimensional manifolds, and inputs arising from reciprocal connections between the MEC and the hippocampus that coordinate their joint representation of position.

We first examine whether such an architecture can enforce a coordinated representation of position in both brain regions. The joint system must be able to represent all possible positions within each environment by coordinated and persistent patterns of neural activity. To achieve this goal, while accounting for the global remapping of place cells (*Muller and Kubie, 1987*) and the phase shift of grid cells (*Fyhn et al., 2007*) across distinct environments (*Jeffery, 2011*), we evoke theoretical ideas on the representation of multiple spatial maps by attractor dynamics in the hippocampus (*Battaglia and Treves, 1998*; *Monasson and Rosay, 2013*; *Samsonovich and McNaughton, 1997*), and extend them to construct synaptic connectivity that supports a representation of position jointly with the grid cell system.

We demonstrate that the coupling between hippocampal and entorhinal attractor networks enables two computational functions. First, integration of velocity inputs in the grid cell network drives similar updates of the place cell representation of position. Idiothetic path integration can thus be implemented within the grid cell system, using fixed synaptic connectivity that does not require readjustment in new environments. Second, the reciprocal connectivity between grid cells and place cells yields an error correcting mechanism that eliminates incompatible drifts in the positions represented by distinct grid cell modules, independently from sensory inputs. Such independent drifts would otherwise rapidly lead to catastrophic readout errors of the grid cell representation (*Burak, 2014*; *Fiete et al., 2008*; *Mosheiff and Burak, 2019*; *Sreenivasan and Fiete, 2011*; *Welinder et al., 2008*) and would therefore be highly detrimental for the coding of position by grid cell activity.

Next, we show that our model exhibits two emergent features that provide a compelling explanation for several recent experimental observations.

## Variability of individual grid cell firing rates across firing fields

Even though grid cell activity is spatially periodic to a good approximation, several recent works showed that firing rates of individual grid cells vary across firing fields (*Diehl et al., 2017*; *Dunn et al., 2017*; *Ismakov et al., 2017*). These firing rate differences are retained across multiple exposures to the same environment. On their own, attractor models of grid cell dynamics predict that firing rates should be identical across firing fields. However, external inputs to the grid cell system could modulate the activity of grid cells in a non-periodic manner. Such inputs could have a sensory origin and project to grid cells from the LEC or the hippocampus. Here, we show that even without any sensory inputs projecting into the system, the embedding of multiple spatial maps within the hippocampus, combined with the synaptic projections from place cell to grid cells, generates variability in the firing rate of grid cells across firing fields.

## Artificial hippocampal remapping upon grid cell depolarization

Place cell firing fields were recently recorded during hyperpolarization or depolarization of layer II stellate cells in the MEC (*Kanter et al., 2017*), giving rise to surprising observations. First, depolarization in the MEC led to a form of remapping in the hippocampus that resembles partial remapping (*Colgin et al., 2008*; *Latuske et al., 2017*; *Muller and Kubie, 1987*; *Quirk et al., 1990*). Some place cells shifted their firing fields to new locations, while other place fields remained anchored to their

original positions and exhibited only rate remapping. This implies that in contrast to global remapping, population patterns of activity under depolarization of the MEC retained partial overlap with firing patterns that were expressed under non-perturbed conditions. Unlike classical remapping experiments, the changes in the firing patterns of place cells in *Kanter et al., 2017* were induced without any changes in the sensory inputs. We refer to this form of remapping induced by MEC depolarization as *artificial remapping*, following *Kanter et al., 2017*. Second, an asymmetry was observed between consequences of depolarization and hyperpolarization in the MEC, since hyperpolarization had only mild effect on the spatial tuning curves of place cells. So far, these observations have lacked a satisfying theoretical explanation. We show below that our model exhibits the same phenomenology as observed in *Kanter et al., 2017*. The theory predicts that MEC depolarization, but not hyperpolarization, alters the manifold of persistent population activity patterns, such that the hippocampus expresses mixtures of patterns that were associated with different environments before the perturbation. The emergence of such mixed states under certain perturbations is characteristic of models of associative memory. Thus, artificial remapping may offer the first experimental observation of this phenomenon in a well-characterized neural network.

## Results

We first ask whether bidirectional connectivity between grid cells and place cells can enforce persistent states of activity that enable continuous coding of position, and are compatible with the observed phenomenology of the place and grid cell neural codes in multiple environments: global hippocampal remapping (*Muller and Kubie, 1987*), and phase shifts in individual modules of the MEC (*Fyhn et al., 2007*). For simplicity, and to reduce the computational cost of simulations, we consider throughout this manuscript a one-dimensional (1d) analogue of grid cell and place cell dynamics.

### Model architecture and description

We model neural activity using a standard rate model. Neurons are grouped into several sub-populations: hippocampal place cells and three grid-cell modules. The architecture of synaptic connections is briefly described here, and fully specified in Methods. There are three types of synaptic connections in the model:

### Recurrent connectivity between grid cells

Connectivity within each grid cell module produces a continuous attractor with a periodic manifold of steady states. We adopt the simple synaptic architecture proposed in *Guanella et al., 2007*. In the 1d analogue, this form of connectivity maps into the ring attractor model (*Ben-Yishai et al., 1995*; *Redish et al., 1996*; *Skaggs et al., 1995*; *Zhang, 1996*): neurons, functionally arranged on a ring, excite nearby neighbors, while global inhibitory connections imply that distant neurons inhibit each other. This synaptic architecture leads to coactivation of a localized group of neurons, which can be positioned anywhere along the ring. The possible positions of the bump are associated with the 1d spatial coordinate by tiling them periodically along the spatial dimension, with an environment specific phase, in accordance with the experimentally observed phase-shifts of grid cell responses under global remapping (*Fyhn et al., 2007*).

### Recurrent connectivity between place cells

Connectivity between place cells is based on similar principles, with two differences: first, each population activity pattern is mapped to a single spatial location. Second, the network possesses a discrete set of continuous attractors, each mapping a distinct environment to a low dimensional manifold of population activity patterns. The synaptic connectivity can be expressed as a sum over contributions from different maps, in similarity to the sum of contributions associated with discrete memories in the Hopfield model (*Hopfield, 1982*). To mimic the features of global remapping, spatial relationships between place cells in the different maps are related to each other by a random permutation. This is similar to previous models (*Battaglia and Treves, 1998*; *Monasson and Rosay, 2013*; *Samsonovich and McNaughton, 1997*) but adapted here to the formalism of a dynamical rate model.

## Connectivity between grid cells and place cells

We first define an idealized tuning curve for each grid cell and each place cell, in each environment. These tuning curves are determined by the mapping between attractor states and positions, as described above and in Methods. For each pair of grid and place cells, we evaluate the correlation between their spatial tuning patterns across all spatial positions and environments. The efficacy of the synapse is then linearly related to this correlation coefficient.

## Coordinated persistent representations

We demonstrate that coordinated persistent representations, and only them, simultaneously co-exist in grid cells and place cells. Our methodology is based on quantitative mapping of persistent states obtained when starting from multiple types of initial conditions. We require that in all cases the system will settle on a persistent activity bump in one of the hippocampal maps, and on persistent activity bumps in each grid cell module at matching locations. Examples of such states are shown in *Figure 2*. Each state exhibits unimodal and co-localized activity bumps when cells are ordered according to their firing location in a particular environment, whereas scattered place cell activity accompanied with grid cell realignment is observed when cells are ordered according to their firing location in other environments.

To test whether a persistent state represents coherently a valid position in one of the environments, we define a *bump score* that quantifies the existence of a unimodal activity bump in each sub-network. In the hippocampal sub-network, a high bump score should be seen only in a single spatial map. Within that map, we also measure the spatial location that corresponds to activity bumps in each sub-network (Methods). This measurement allows us to verify that all sub-networks express activity bumps in compatible positions, and to test the persistence of these bump states. *Figure 3* shows the results from several types of initial conditions:

1. 'Consistent condition': Grid cell and place cell activities are initially set to encode an identical location within the same map. Using these initial conditions, we verify that all locations, in all environments, can be represented by persistent states (*Figure 3a–d*). The bump score remains high after 1 s when initializing the network in co-localized bump states, and the drift of the represented position is small compared to the width of activity bumps, indicating that these states are persistent.
2. 'Inconsistent condition': Grid cell and place cell activities are initially set to encode different locations. Using this test, we verify that such incompatible states are unstable and give rise (after a short transient period) to a consistent state (*Figure 3e–h*).
3. 'All random': Both grid cell and place cell activities are set to random initial rates. The goal of these tests is to verify that even if bump states do not exist initially, the system evolves into unimodal activity bumps in compatible positions, and that no other spurious states emerge. As expected, co-localized place cell and grid cell activity bumps appear with equal probabilities across the different maps (*Figure 3i*).
4. Finally, we test how the system evolves from two additional types of initial states: in one condition grid cells are initially set to have random rates, while place cells are set to encode a specific location. In the second condition, grid cells are set to encode a specific location, and place cells are assigned with random rates. The results (*Figure 3—figure supplement 1*) indicate that both place cells and grid cells are capable of attracting activity in the other network to organize in a compatible manner with their initial state (note however that in the 'inconsistent' initial condition, under our choice of coupling parameters, the final locations of persistent states are typically determined by the initial place cell location).

*Figure 3—videos 1–3* show examples of activity dynamics in the network for initial conditions of types (1–3) and *Figure 3—figure supplement 2* quantify the dynamics of the bump score for each type of the initial conditions shown in *Figure 3*.

## Functional consequences: path integration and dynamic coupling

Two functional consequences arise from the coupling between the place and grid cell networks:

### Path integration

The MEC has been hypothesized to be the brain area responsible for idiothetic path integration for several reasons (*Burak, 2014*; *Hafting et al., 2005*; *McNaughton et al., 2006*) among them the

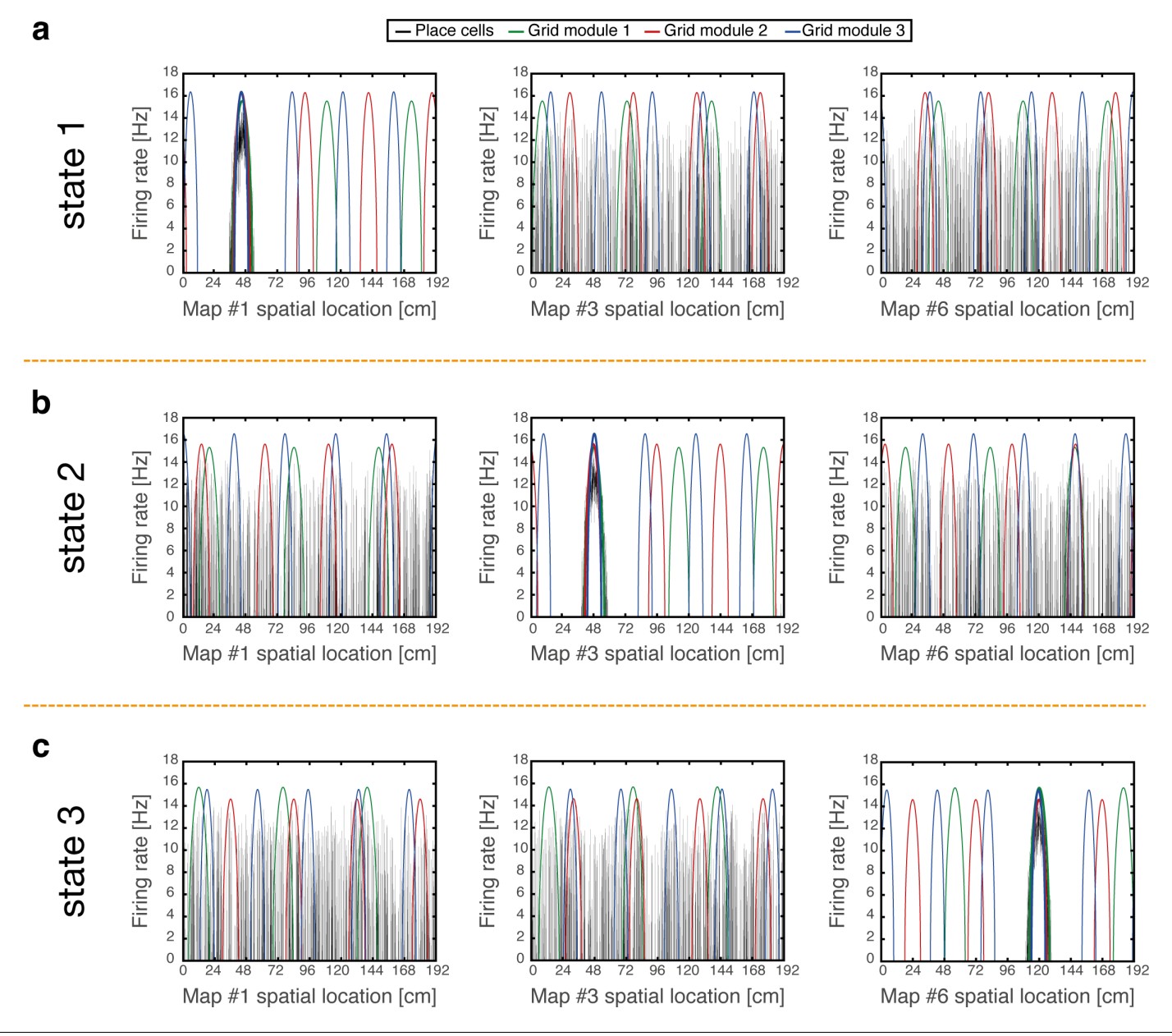

**Figure 2.** Examples of simulation results, demonstrating joint persistent states of place cells and grid cells. (a) One persistent state of the network (state 1). In each panel, cells are ordered according to their preferred firing locations in one environment. In each grid cell module, positions displaced by any integer multiple of the grid spacing correspond to the same cell, whereas distinct place cells span the whole extent of the environment. When cells are ordered according to their preferred firing locations in map #1 (left), a unimodal place cell bump is observed, and the distinct grid cell bumps of the three modules co-localize around the same spatial location. When the cells are ordered according to their preferred firing locations in a different map (middle - map *3* and right - map *6*), the same place cell activities are scattered throughout the environment and do not resemble a unimodal activity bump. Furthermore, grid cell bumps do not necessarily co-localize around the same spatial location, as they independently realign. (b-c) Examples of persistent states (state 2 and 3), similar to (a) but representing positions in map *3* and in map *6* respectively. Note that here the activity of place cells seems scattered and grid cell bumps independently realign when cells are ordered according to their preferred firing locations in map #1.

highly geometric nature of grid cell spatial selectivity, the existence of inputs to the MEC that convey information about the animal's self-motion and heading (*Kropff et al., 2015*; *Sargolini et al., 2006*), and the impairment of path integration following disruption of grid cell activity (*Gil et al., 2018*). The similarity in response patterns of grid cells across environments implies that implementation of path integration in the MEC could be accomplished in different environments by the same synaptic

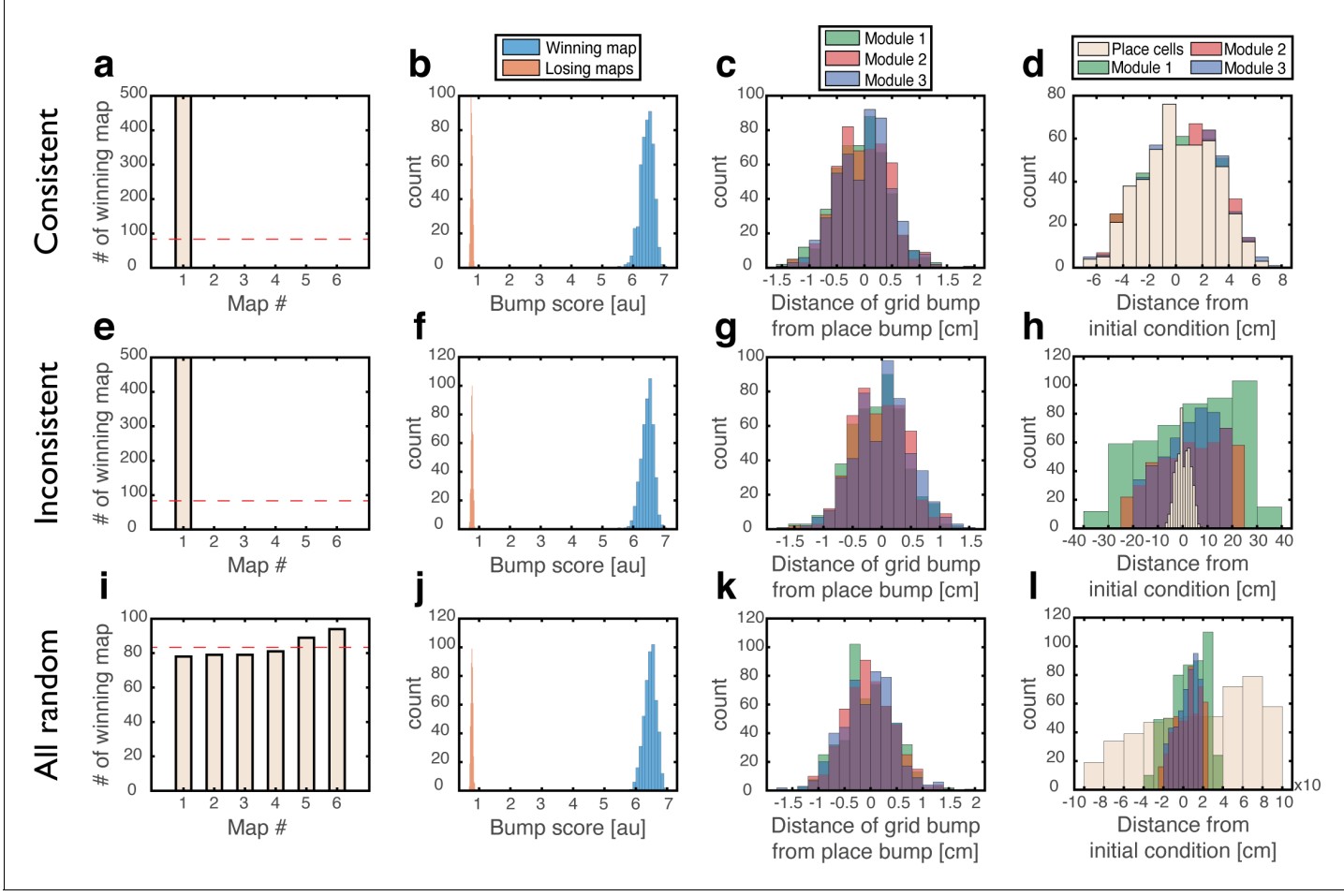

**Figure 3.** Quantitative mapping of the persistent states expressed by the joint network. In all panels, the system is placed at 500 initial conditions of three types (**a-d**: 'consistent', **e-h**: 'inconsistent', **i-l**: 'all random'). Its state is then analyzed after a 1 s delay period. (**a-d**) Results from the 'consistent' initial condition (the system is placed without loss of generality in map #1). (**a**) Histogram of winning map, defined as the map that achieved the highest bump score. Red dashed line: uniform distribution. (**b**) Histogram of place cell bump scores, obtained from the winning map (blue) and from an average on all other maps ('Losing maps', orange). (**c**) Histogram of distances between the measured spatial location of each grid cell bump and the place cell bump, evaluated in the coordinates of the winning map. (**d**) Histogram of distances traveled by the activity bumps from their initial condition, evaluated in the coordinates of the winning map. (**e-h**) Similar to (**a-d**) but for 'inconsistent' condition. (**i-l**) Similar to (**a-d**) but for 'all random' condition. The initial position was evaluated using the same algorithm used in (**d,h**) but in this case, where there were no initial bumps, the outcome depended on slight random variations in the initial condition and thus, effectively, the initial position was chosen randomly. Note that for all initial conditions the state of the system after the delay period corresponded to a single spatial map (**b,f,j**), and the place cell and grid cell representations are coordinated (**c,g,k**). The online version of this article includes the following video and figure supplement(s) for figure 3:

**Figure supplement 1.** Analysis of persistent states following initial conditions of type (4).
**Figure supplement 2.** Bump score dynamics of persistent states.
**Figure 3—video 1.** Dynamics under 'consistent' initial condition.
https://elifesciences.org/articles/56894#fig3video1
**Figure 3—video 2.** Dynamics under 'inconsistent' initial condition.
https://elifesciences.org/articles/56894#fig3video2
**Figure 3—video 3.** Dynamics under 'all random' initial condition.
https://elifesciences.org/articles/56894#fig3video3

connectivity, whereas implementation in the hippocampus (*McNaughton et al., 1996*) would require establishment of dedicated synaptic connectivity in each environment. *Figure 4a* demonstrates that in the coupled system, integration of velocity inputs in the MEC, using mechanisms described in *Burak and Fiete, 2009* (Methods) induces coordinated updates in the position represented in the place cell network and *Figure 4—figure supplement 1* shows the dynamics of the corresponding

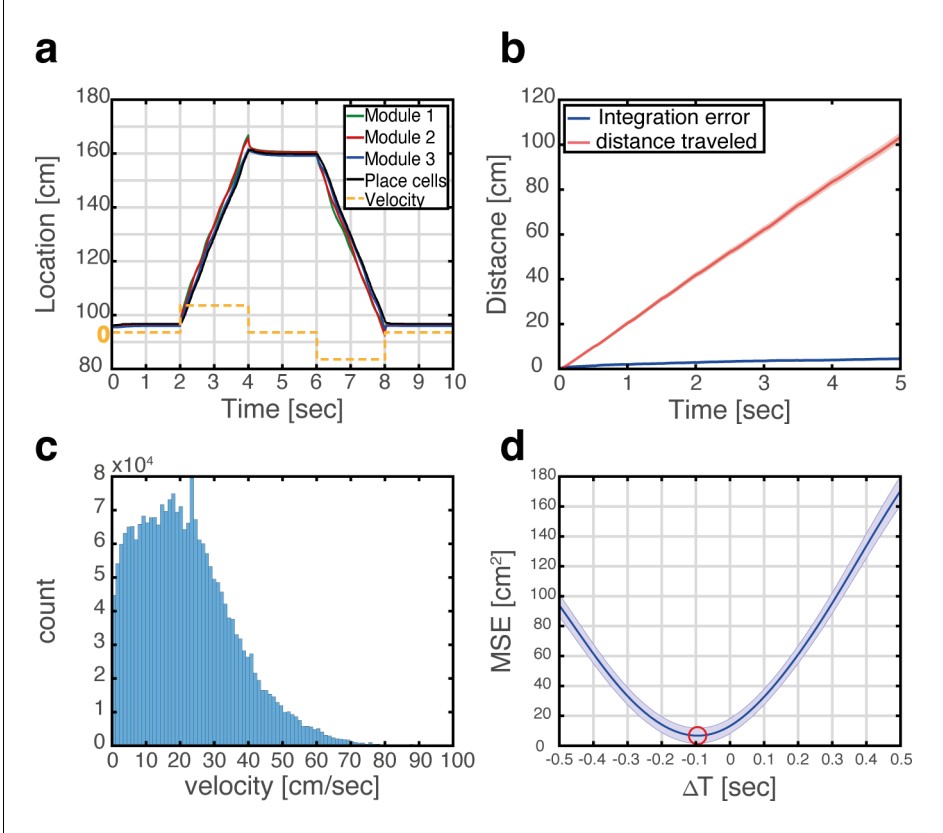

**Figure 4.** Velocity integration in grid cells can update the place cell representation. (**a**) Readout of place cell bump location (black) while grid cell modules (green, red, and blue) integrate a velocity profile (dashed orange line). (**b**) Mean absolute distance between measured location of the place cell activity bump, and integrated velocity (blue), as a function of time. This quantity is compared with the total distance traveled by the place cell bump (red), which is much larger. Averaging was performed over 500 trajectories. In each trajectory velocity was randomly sampled every 0.5 s from an experimentally measured velocity distribution, which was obtained during foraging of a rat in an open-field environment (see panel c). Shaded error bars are 1.96 times the standard deviations obtained from each simulation, divided by the square root of the number of realizations (corresponding to a confidence interval of 95%). (**c**) Histogram of velocities measured experimentally during foraging of a rat in an open field environment (*Hafting et al., 2005*). (**d**) Mean squared displacement between positions represented by place cells and grid cells, when evaluated at varying time lags between the two measurements. Red circle marks the minimal MSE. The minimum MSE occurs at a negative time lag (~−100 ms), which indicates that the place cell representation lags behind the grid cell representation. Shaded error bars are 1.96 times the standard deviations obtained from each simulation, divided by the square root of the number of realizations (corresponding to a confidence interval of 95%).

The online version of this article includes the following figure supplement(s) for figure 4:

**Figure supplement 1.** Bump score dynamics for each of the embedded maps, corresponding to the realization shown in *Figure 4a*.

bump scores across multiple maps. *Figure 4b* shows that errors accrued during path integration remain small, when using realistic velocities that were measured experimentally during foraging in an open field environment (*Figure 4c*). Interestingly, the representation in the hippocampus lags behind the entorhinal representation, due to synaptic transmission delays (*Figure 4d*).

## Coordination of the grid cell representation: suppression of incompatible drifts

The modular structure of the grid cell code for position confers it with large representational capacity (*Fiete et al., 2008*). Yet, the modularity poses a significant challenge for the neural circuitry that

maintains the representation and updates it based on self-motion. The configurational space of *M* grid module phases is *M* dimensional in 1-d (or 2*M* dimensional in 2-d), but phases that correspond to continuous motion in a given environment span a much smaller subspace: during continuous motion, changes in the phases of any two modules should be identical, up to a proportionality factor related to their respective grid spacings. However, in the absence of correcting mechanisms phase errors that accrue in different modules due to noise might not adhere to these restrictions. Such incompatible drifts can rapidly lead to combinations of phases that do not represent any position in the nearby vicinity of the animal, resulting in catastrophic readout errors (*Figure 5a*) (see also *Burak, 2014*; *Sreenivasan and Fiete, 2011*; *Welinder et al., 2008*).

Sensory cues may correct drifts in individual modules and restore the coordination between the phases of different modules. It has been shown that such phase resets occur during encounters with walls, possibly mediated by an interaction between grid cells and border cells (*Hardcastle et al., 2015*; *Mulas et al., 2016*; *Pollock et al., 2018*). However, it has been argued that under conditions in which sensory inputs are absent or poor, other mechanisms must act to coordinate grid cell phases and prevent catastrophic readout errors (*Burak, 2014*; *Fiete et al., 2008*). Two possible mechanisms have been proposed: in one solution (*Sreenivasan and Fiete, 2011*; *Welinder et al., 2008*), the hippocampal network reads out the position jointly represented by grid cell modules, and projections from hippocampus to the MEC correct small incompatible drifts accrued in the different modules. A second solution, proposed more recently (*Mosheiff and Burak, 2019*) involves synaptic connectivity between grid cells belonging to different modules. The first solution has been explored previously (*Sreenivasan and Fiete, 2011*) without explicit modeling of attractor dynamics in the hippocampus, and without considering the embedding of multiple spatial maps in the entorhinal-hippocampal network. Therefore, we tested whether coordination of modules can be implemented in the coupled system, across multiple environments.

We first considered the consequences of stochastic neural firing, by modeling grid cells as noisy (Poisson) units. In the absence of coupling between the MEC and hippocampus the positions represented in different modules drifted randomly in an uncoordinated manner (*Figure 5b*, top). With bidirectional coupling (*Figure 5b*, middle) the representation of position accrued errors relative to the true position of the animal, but remained coordinated in all modules and in the hippocampus. *Figure 5b* (bottom) quantifies the relative drift between modules in the different conditions over multiple realizations of the dynamics, showing that drifts remain coordinated under coupling with the hippocampus. Drifts that are coordinated across modules are much less detrimental for decoding of position than coordinated drifts [see (*Burak, 2014*; *Mosheiff and Burak, 2019*) for a thorough discussion].

Next, we demonstrated that the system is resilient to noise in the velocity inputs to the different modules (*Figure 5c*). Even in an extreme situation, in which modules received highly incompatible velocity inputs, their motion remained coordinated in the coupled network but not in the uncoupled network.

## Emergent properties in the hippocampus and MEC

The embedding of multiple spatial maps in the coupled hippocampal-entorhinal network gives rise to two non-trivial emergent properties.

### Variability of individual grid cell firing rates across firing fields

As the animal moves in a given environment, the summed synaptic input from all place cells to a given grid cell includes a contribution associated with that environment, which is spatially periodic. However, grid cell synaptic inputs also receive contributions from place cells associated with other spatial maps that are embedded in the synaptic connectivity. The latter contributions are spatially aperiodic within the present environment, leading to variability across fields. As an example, *Figure 6a* shows the population activity of place cells and grid cells from module #3. The system is placed at three spatial locations that are displaced from each other by the grid spacing of module #3. Thus, the same grid cells are active in these three locations. However, the amplitude of the grid cell activity bump differ at the three locations. *Figure 6b* quantifies the variability of peak firing rates generated by individual grid cells across firing fields within a single environment, using the coefficient of variation (CV): a histogram of the CV, collected from all cells, is shown for networks with

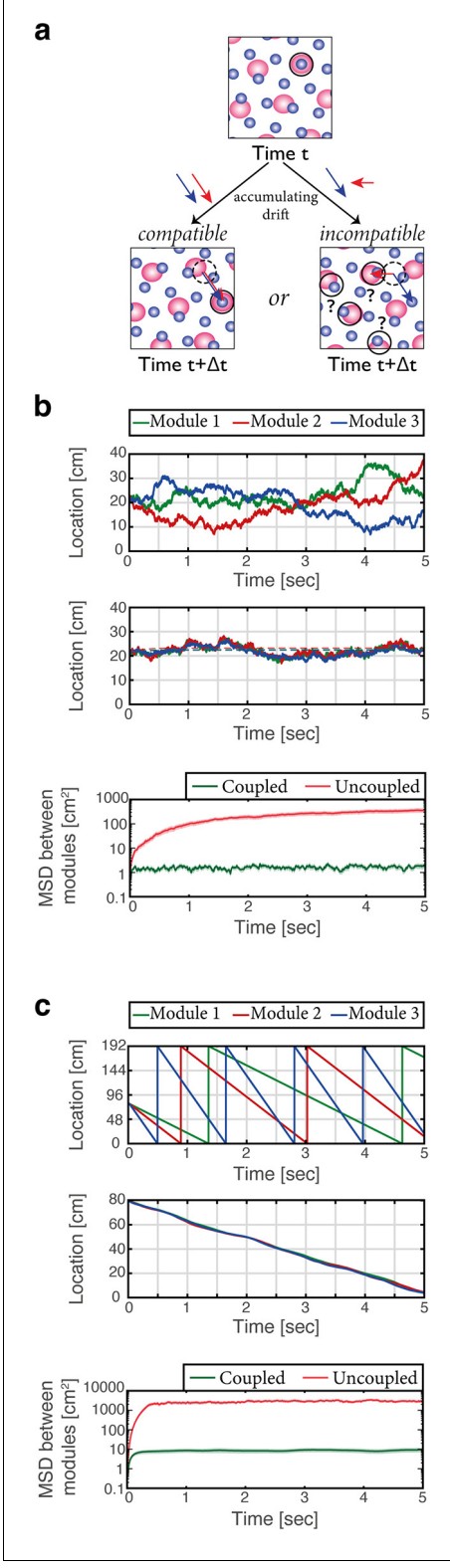

**Figure 5.** Place cells coordinate the representations of position in distinct grid cell modules. (a) The need for coupling. Top panel: blue- and red-shaded areas represent schematically the posterior likelihood for the animal's position in 2-d, obtained from the activity of *Figure 5 continued on next page*

varying number of embedded maps. CVs vanish when a single spatial map is embedded in the synaptic connectivity and increase with the addition of spatial maps.

## Artificial hippocampal remapping upon grid cell depolarization

In a recent experiment (*Kanter et al., 2017*), MEC layer 2 stellate cells were reversibly hyperpolarized or depolarized using chemogenetic receptors, while animals were foraging in a familiar environment. Chronic tetrode arrays implanted in the hippocampus were used to monitor CA1 firing fields during these manipulations. Hyperpolarization of the MEC had very little effect on place cell firing fields, whereas depolarization of the MEC elicited dramatic effects: a significant portion of place cells exhibited changes in the locations or rates of their firing fields, resembling those seen during global and rate remapping experiments. This effect included place cells that had been turned off or on as seen in natural remapping experiments. Firing fields of all other cells remained in their baseline positions, without significant changes. As for the grid cells, both manipulations led to changes in the firing rates but did not elicit a change in the firing locations, unlike the coherent shifts and rotations seen in distinct familiar environments (*Fyhn et al., 2007*).

The asymmetric effect of entorhinal depolarization and hyperpolarization was highlighted (*Kanter et al., 2017*) as the most puzzling experimental outcome. Indeed, under simple feedforward models of place cell emergence from grid cell inputs (*de Almeida et al., 2009*; *Monaco et al., 2011*; *Neher et al., 2017*; *Rolls et al., 2006*; *Solstad et al., 2006*), it is difficult to envision why such an asymmetry would be observed. Overall, a theoretical framework that can coherently explain the consequences of depolarization and hyperpolarization of the MEC hasn't been available so far.

To test the effect of grid cell depolarization or hyperpolarization in our model, we adjusted the excitability of grid cells by adding a fixed contribution to their synaptic drive, which was either excitatory or inhibitory. We first placed the system in persistent states in one map (map 1). We then hyperpolarized the grid cell network. Place cell bumps remained around their baseline spatial location, and activity in the place-cell network exhibited significant overlap only with map 1 (*Figure 7a–b*). Similar results from control trials, in which no manipulation was performed, are

*Figure 5 continued*

all grid cells in module 1 (red) and module 2 (blue). Periodic colored blobs correspond to areas with high likelihood for the position, while blank areas correspond to low likelihood (note that since the blobs represent a decoded likelihood from the activity of all grid cells in the same module, they should not be confused with the spatial receptive fields of single cells). The position which is most likely given the joint activity in both modules is designated by a black circle. Bottom left: when noise-driven drifts in the two modules are identical (compatible drifts, blue and red arrows), the outcome is a shift in the represented position (original location marked by black dashed circle for reference). Bottom right: incompatible drifts in the two modules may result in abrupt jumps in the maximum-likelihood location (*Burak, 2014*; *Mosheiff and Burak, 2019*). (b) Examples of 1-d simulation results starting from a consistent initial condition showing grid cell module bump locations (green, red and blue) vs. time without (up) and with (middle) coupling between place cells and grid cells. Random accumulated drifts are driven by intrinsic neural noise, modeled as arising from Poisson spiking. Dashed lines in middle panel are generated from a non-noisy simulation for reference. Bottom: mean square displacement (MSD) between bump locations of distinct grid cell modules (averaged across all pairs and realizations of the stochastic dynamics) vs. time with (green) and without (red) coupling between place cells and grid cells, starting from a consistent initial condition. Shaded error bars are 1.96 times the standard deviations obtained from each simulation, divided by the square root of the number of realizations (corresponding to a confidence interval of 95%). (c) Similar examples (top, middle) and analysis (bottom) as in (b) but obtained from simulations in which drifts arise from incompatible velocity inputs provided to the distinct modules.

shown in *Figure 3a–b*. The minor effect of grid cell hyperpolarization is not surprising, since the hippocampal network is structured such that it can sustain population activity patterns associated with each environment even without inputs from grid cells. Therefore, suppression of activity in the MEC network did not significantly alter the population activity patterns expressed by the place cell network.

In contrast with this result, depolarization of the MEC network elicited significant changes in the structure of population activity patterns expressed by the hippocampal network. Following a short transient, population activity of place cells stabilized on patterns that exhibited overlap with patterns that were previously associated with multiple spatial maps (*Figure 7c*). The overlap was highly significant compared to overlaps obtained with random maps that were not embedded in the connectivity (gray bars in *Figure 7c*). Thus, the place cell network expressed mixtures of population activity patterns from different maps stored in the hippocampal connectivity. Typically, the highest overlap remained with map 1 (*Figure 7d*). An example of population activity patterns before and after depolarization in one location is shown in *Figure 7—figure supplement 1a–d*. (Corresponding dynamics can be seen in *Figure 7—video 1*, and *Figure 7—figure supplement 2* quantifies the bump score dynamics, from all realizations shown in *Figure 7*.) In similarity to experimental results (*Kanter et al., 2017*), both depolarization and hyperpolarization changed the firing rate, but not firing location, of MEC neurons. On average, grid cells increased their firing rates under grid cell depolarization and decreased their firing rates under grid cell hyperpolarization (not

shown), in agreement with the experiment. *Figure 7—figure supplement 1e* demonstrates that the mixed activity patterns are persistent.

Our interpretation of artificial remapping is thus that excessive inputs from grid cells enhance the quenched noise arising from multiple embedded maps, and that this enhancement puts the system in a regime in which it expresses mixed states. To demonstrate the existence of such a regime, we show in *Figure 7—figure supplement 3a–b* that similar expression of mixed states emerges when synapses between grid and place cell are strengthened, even without grid cell depolarization. In contrast, selective strengthening only of the component associated with one of the maps, does not generate mixed states (*Figure 7—figure supplement 3c–d*). On the contrary: this selective enhancement suppresses the emergence of mixed states under grid cell depolarization (*Figure 7—figure supplement 3e–f*). These results demonstrate that the emergence of mixed states is not simply due to enhancement of inputs from the MEC, but is due to the amplification of quenched noise.

To make direct contact with the experimental observations (*Kanter et al., 2017*), we examined the firing properties of individual place cells as a function of the animal's position during locomotion under MEC depolarization. The observation that population activity patterns under depolarization are mixtures of activity patterns from different maps led us to expect that some cells would maintain place fields in their baseline locations, while others would remap. To test this hypothesis, we first

placed the system in a state corresponding to a particular position in map 1. Then, while applying the perturbation, we induced path integration in the MEC network and monitored population activity patterns while continuously scanning positions along the environment.

In accordance with the hypothesis, we found that cells exhibited the same phenomenology observed experimentally (*Kanter et al., 2017*; *Figure 8a–g*). Many cells shifted their firing location (~29%), whereas others (~18%) maintained their place fields in their baseline location. Of these, some cells exhibited rate remapping (~7%), while the firing rate of other cells remained unaffected (~11%). Some place cells expressed one or (rarely) more firing fields in addition to their original field location (~36.5%), some completely turned-off (~6%) or exhibited a minor field (~1.5%), and some place cells remapped into more than a single field (~9%). The percentages of place cell responses are likely influenced by the model parameters (*Table 1*), and depend also on the precise classification criteria (Methods). Qualitatively, however, all the response types observed in our model have been observed experimentally (*Kanter et al., 2017*). All active cells exhibited a continuous activity pattern with one, or occasionally few, unimodal firing fields. *Figure 8—figure supplement 1* shows all the population activity patterns observed while traversing the environment in a single path integration cycle. In agreement with the experimental observations, firing patterns returned to baseline configurations following reversal of MEC depolarization (not shown). As expected (*Figure 7a–b*), hyperpolarization did not elicit significant changes in the firing locations of place cells (*Figure 8h*).

Finally, we examined how grid cell activity is influenced by the altered firing patterns of place cells during grid cell depolarization. Our previous observation that place cell inputs induce variability in firing rates of single grid cells across different firing fields, led us to examine whether the rank order of grid fields according to peak firing changes under the modification of place cell firing patterns associated with grid cell depolarization (similar rank changes have been observed also under place cell rate remapping, but in this case changes were induced by modifications of sensory inputs in a familiar environment [*Diehl et al., 2017*]). *Figure 8—figure supplement 2* demonstrates that when a single spatial map is embedded in the synaptic connectivity the rank correlation is equal to unity, as expected. When multiple spatial maps are embedded in the connectivity, the rank order correlation weakens, in accordance with experimental observations (*Kanter et al., 2017*).

## Discussion

We demonstrated how multiple attractor networks in the hippocampus and MEC can be coupled to each other, such that they jointly and coherently represent positions across multiple spatial maps,

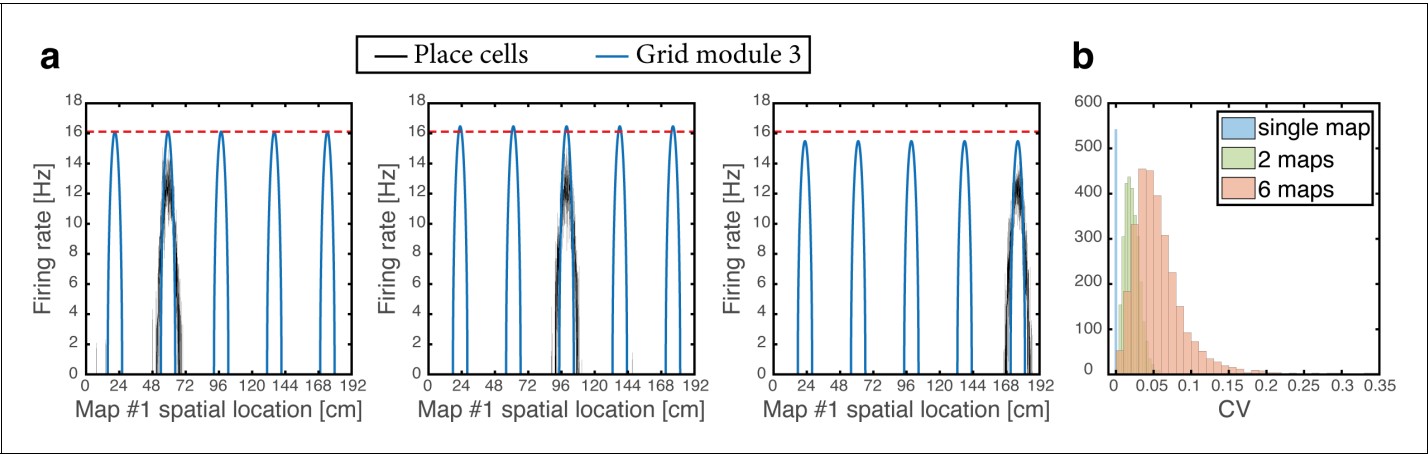

**Figure 6.** Spontaneously emerging variability of individual grid cell firing rates across firing fields. (**a**) Simultaneous firing rates of place cells (black) and grid cells from one module (module #3, blue), shown at three different persistent states that represent periodic locations of module #3. Even though the same grid cells are active in all three locations, the amplitude of the grid cell activity bumps differ (compared with the red dashed line which shows the maximal grid cell firing rate achieved at the left example, for reference). Different place cells are active in the three examples, thus providing different quenched noise to the same grid cells in each case. (**b**) Histogram showing CV of maximal firing rate obtained from all individual grid cells across firing fields within a single environment, shown for networks with a single map (blue), two maps (green), and six maps (red).

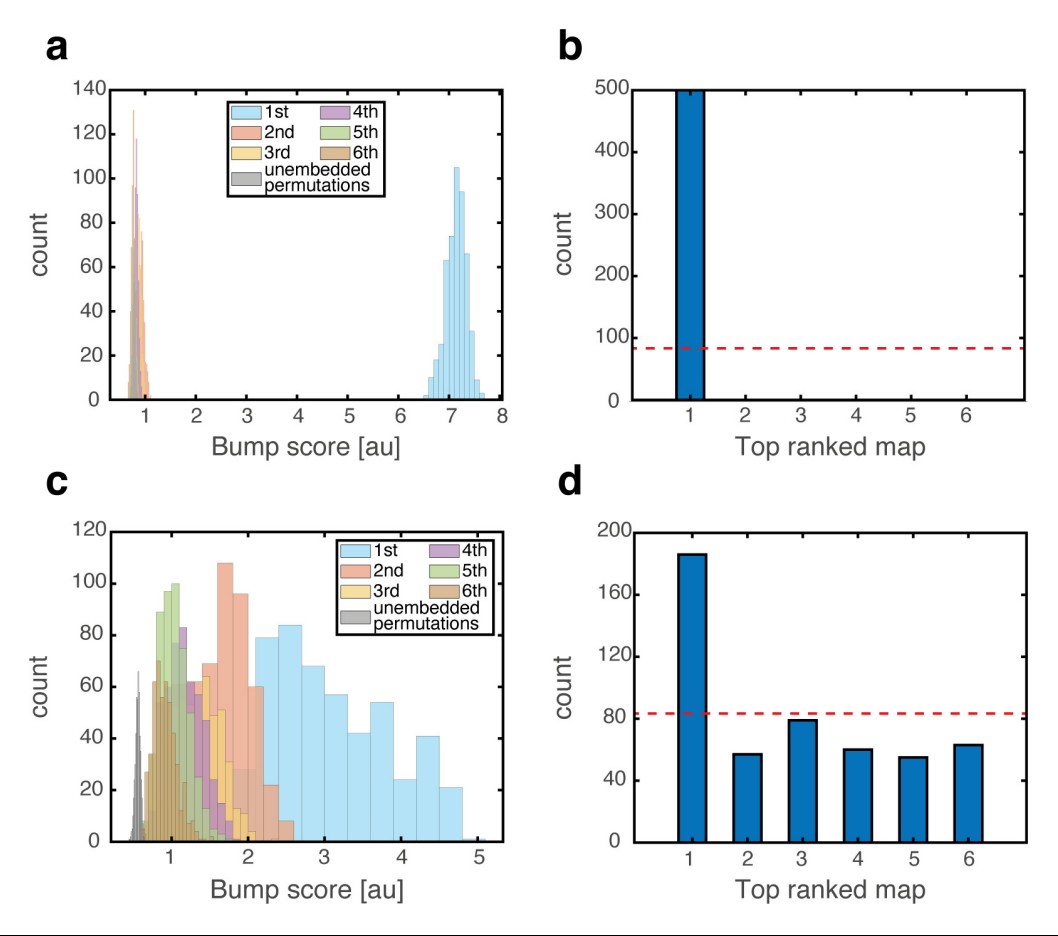

**Figure 7.** Emergence of persistent mixed states under depolarization but not under hyperpolarization of grid cells. (a) Histogram showing bump score distributions for all embedded maps under grid cell hyperpolarization. In each simulation, the system is placed in a 'consistent' initial condition, and hyperpolarization is applied. Spatial maps are then ranked according to their bump score. Each color corresponds to a histogram over all bump scores from maps with specific rank, regardless of the map's identity (this is identical to the analysis shown in *Figure 3b,g,f* except that here losing maps are separated into distinct histograms according to rank). Bump scores for random unembedded maps are shown in gray. (b) Distribution of identities of the top scored maps from (a). Dashed red line shows the uniform distribution. In all realizations, top ranked map is map #1. (c) Same as (a) under grid cell depolarization. Bump scores are lower than the winning scores in *Figure 3b*. Bump score distributions from differently ranked maps overlap, and all histograms exhibit significantly higher bump scores than those of unembedded maps (gray), indicating simultaneous expression of activity patterns from multiple embedded maps. (d) Same as (b) under grid cell depolarization.

The online version of this article includes the following video and figure supplement(s) for figure 7:

**Figure supplement 1.** Persistent mixed state examples, and maintenance of grid cell firing location but not firing rate under grid cell depolarization.

**Figure supplement 2.** Bump score dynamics under grid cell hyperpolarization and depolarization.

**Figure supplement 3.** Mixed states can be induced or suppressed by changing the connectivity between grid cells and place cells.

**Figure 7—video 1.** Dynamics under grid cell depolarization.

https://elifesciences.org/articles/56894#fig7video1

while exhibiting the features of global remapping that are experimentally observed in both brain areas.

Due to the embedding of multiple spatial maps in the hippocampal network, the input to a place cell in a given environment can be separated into two components. The first component arises from the synaptic connectivity associated with the presently expressed spatial map. The other component

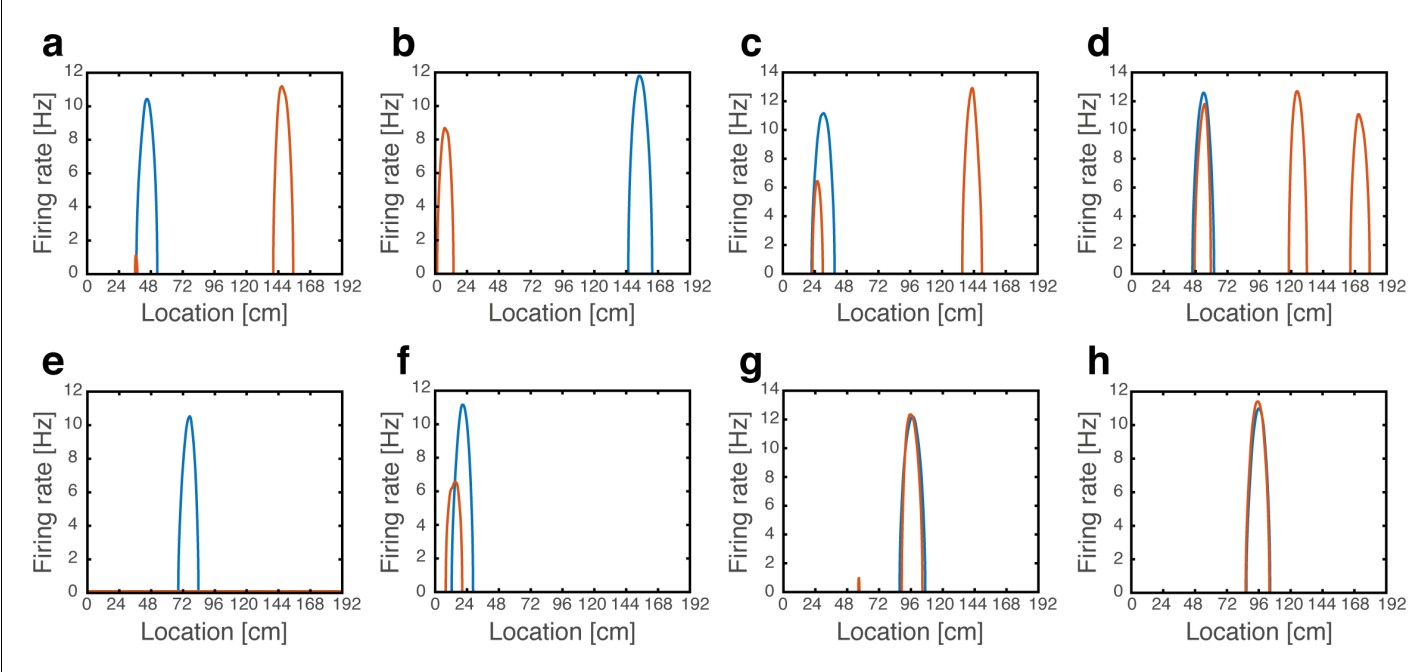

**Figure 8.** Artificial remapping of place cells following depolarization, but not hyperpolarization of grid cells. (**a-g**) Firing rate maps of seven representative place cells during locomotion without (blue) and with (orange) grid cell depolarization. Rate maps show that under grid cell depolarization place cells changed locations (**a-b**), acquired additional fields (**c-d**), turned off (**e**), rate remapped (**f**) or were unaffected (**g**). (**h**) Rate maps of a representative place cell without (blue) and with (orange) grid cell hyperpolarization.

The online version of this article includes the following figure supplement(s) for figure 8:

**Figure supplement 1.** Place cell population activity patterns observed while traversing the environment using path integration.

**Figure supplement 2.** Altered firing rate relationship between individual grid fields is observed during grid cell depolarization.

is associated with other spatial maps, and contributes quenched noise to the synaptic inputs (*Battaglia and Treves, 1998*; *Monasson and Rosay, 2013*; *Samsonovich and McNaughton, 1997*) (Appendix 1). Due to this quenched noise, the system loses its ability to represent all spatial positions in true steady states of the network dynamics. However, if the quenched noise is sufficiently weak, bump states representing all possible positions remain highly persistent. This feature of continuous attractor systems with quenched noise, and specifically with multiple maps, has been extensively discussed in previous works (*Itskov et al., 2011*; *Monasson and Rosay, 2014*). For this reason, we tested for the existence of persistent rather than steady states in our analysis (*Figure 3*). In particular, we verified that drifts of the represented location are small, when starting from the consistent condition (*Figure 3d*). The drift rate is expected to be even smaller with a larger number of place cells, since the contribution from place cell inputs to the drift scales inversely with the network size (*Monasson and Rosay, 2014*). Furthermore, it has been demonstrated that sensory inputs, projecting to the hippocampus can pin the representation and prevent drift (*Monasson and Rosay, 2014*).

Systematic drifts (*Itskov et al., 2011*; *Monasson and Rosay, 2014*) and pinning of the attractor state when velocity inputs are small (*Burak and Fiete, 2009*), can reduce the precision of path integration. Yet, *Figure 4b* demonstrates that with a realistic distribution of velocities, the system is still highly effective in integrating the velocity inputs, over multiple spatial maps, even when allocentric sensory inputs are completely absent.

Our model simplified the biological complexity of the hippocampal formation: we did not take into account the laminar organization of the MEC, and the division of the hippocampus into subregions. Our model of the hippocampal network matches most closely CA3, which is believed to underlie the auto-associative features of network activity in the hippocampus, and is the area targeted by layer II of the MEC, which was manipulated by *Kanter et al., 2017*.

**Table 1.**

| Parameter | Value | Units |
|---|---|---|
| $L$ | $6^*$ | none |
| $A$ | $8.31 \cdot 10^{-2}$ | Hz |
| $\sigma$ | 4.8 | cm |
| $h$ | $-2.6 \cdot 10^{-2}$ | Hz |
| $B$ | $75 \cdot 10^{-2}$ | Hz |
| $\rho$ | $\frac{1}{3} \cdot 2\pi$ | radian |
| $k$ | $-6.93 \cdot 10^{-1}$ | Hz |
| $\Delta\theta$ | $\frac{1}{16} \cdot 2\pi$ | radian |
| $\alpha$ | $1.03 \cdot 10^{-2}$ | Hz |
| $\beta$ | $-\frac{20}{3} \cdot 10^{-4}$ | Hz |
| $\tau$ | $15 \cdot 10^{-3}$ | s |
| $\Delta t$ | $2 \cdot 10^{-4}$ | s |
| $\gamma_g$ | 4 | none |
| $I_{pc,0}$ | $-10$ | Hz$^2$ |
| $\gamma_p$ | 50 | none |
| $I_{gc,0}^{\mu}$ | $(-5, -5, -5)$ | Hz$^2$ |
| $\varepsilon^{\mu}$ | $(1.7, 1.9, 2.3)$ | $(cm \cdot s)^{-1}$ |
| $I_{per}^{Depo}$ | 500 | Hz$^2$ |
| $I_{per}^{Hyper}$ | $-100$ | Hz$^2$ |

*In all Figures we simulated the same networks using $L = 6$, except for **Figure 6b** (where $L$ was also set to 1 and 2), and **Figure 8—figure supplement 2** (where $L$ was also set to 1, 2 and 4).

We assumed that the association of place cells to locations in a novel environment is random and occurs jointly with random and independent realignment of grid phases. The connectivity is then set up in order to support the joint representation in the hippocampus and MEC. Under independent remapping in grid cells and place cells, the capacity of the joint representation is limited by the grid-to-place connections (Appendix 1). This result is in line with recent analysis (**Yim et al., 2019**), which indicates that once grid phases are set, realizable patterns of place fields by a binary perceptron (involving a linear weighting of grid cell inputs) are limited. As an alternative to independent remapping in place cells and grid cells, it has been proposed that place cell firing locations may be determined in a new environment by grid cell inputs (**Fyhn et al., 2007**; **Monaco et al., 2011**). We did not explore this scenario explicitly in this work, but in principle, it may be possible to construct a variant of our model in which connections from grid cells to place are kept fixed as new maps are embedded in the network, while the association of place cells to positions in a new environment is determined by these fixed connections, followed by competitive dynamics. The exact mechanism through which spatial fields in novel environments are associated to place cells is not known and since global remapping in place cells can occur also without input from grid cells (**Schlesiger et al., 2018**), it is likely that place fields are not solely determined by grid cells inputs.

Previous theoretical works showed that the grid cell representation has a high dynamic range, which enables coding and storage in working memory of the animal's location with high spatial resolution, over a large range of positions (**Burak, 2014**; **Fiete et al., 2008**; **Mathis et al., 2012**; **Sreenivasan and Fiete, 2011**; **Wei et al., 2015**). However, due to the distributed nature of the grid cell code, it was argued that it is crucial to maintain precise coordination between the positions represented in different grid cell modules (**Burak, 2014**; **Mosheiff and Burak, 2019**; **Sreenivasan and Fiete, 2011**; **Welinder et al., 2008**). Here, we showed that the required coordination can be achieved through bidirectional connectivity between grid cells and place cells, over multiple spatial maps, using an explicit model of attractor dynamics in the MEC and the hippocampus.

The experimentally observed variability in the firing rates of individual grid cells across their firing fields may arise from place cell inputs, as predicted by our model and proposed in other studies (*Dunn et al., 2017*; *Ismakov et al., 2017*). The variability can arise from inhomogeneities in the synaptic connections from place cell to grid cells which are unrelated to the embedding of multiple maps (*Dunn et al., 2017*). Alternately, it can arise from sensory inputs, or from synaptic inputs from different modules (*Ismakov et al., 2017*; *Kang and Balasubramanian, 2019*). Instead, we identified a mechanism which naturally produces this variability, at least in part, simply by the process of embedding multiple spatial maps in the connectivity and is independent from sensory inputs. The fact that the rank order of grid fields was modified under grid cell depolarization when the hippocampal network exhibited artificial remapping (*Kanter et al., 2017*), as observed in our model, provides further support to the hypothesis that inputs from place cells significantly contribute to the variability, independently of sensory inputs.

Our model reproduces the recently observed phenomenology of artificial remapping in the hippocampus under MEC depolarization but not under hyperpolarization. While this phenomenon resembles partial remapping observed under certain manipulations of environmental features (*Colgin et al., 2008*; *Latuske et al., 2017*; *Muller and Kubie, 1987*; *Quirk et al., 1990*), we focused on artificial remapping (*Kanter et al., 2017*) since our interest in this work is in the ability of the MEC and hippocampus to sustain persistent patterns of activity at the population level, irrespective of external inputs. Artificial remapping, as opposed to partial remapping, occurs under a manipulation in which sensory inputs are kept fixed and therefore must arise from mechanisms that are intrinsic to the MEC-hippocampal network.

According to our model, artificial remapping is a consequence of a perturbation to the network that places it in a regime in which it expresses mixtures of population activity patterns. This transition does not require any modification of sensory inputs, in accordance with the experimental protocol (*Kanter et al., 2017*), in which the environment remains fixed. All types of remapping observed experimentally (*Kanter et al., 2017*), emerged in our model except for cells that lacked a firing field before depolarization and acquired one afterwards. This is due to the fact that, for simplicity, we assigned a spatial receptive field to all cells in each environment. It would be straightforward to include in the model silent cells in each spatial map (*Monasson and Rosay, 2013*), in order to obtain cells in the hippocampal network that acquire a firing field only after grid cell depolarization.

Our results also suggest that place cell artificial remapping is likely to occur at specific spatial locations which are determined by the relationship between grid phases and the place cell's firing locations across all maps: remapped place fields are likely to emerge at locations in which the grid phases (associated with the original spatial map) align under the coordinates of another spatial map. Note that artificial remapping occurs only in place cells while grid cell activities do not collectively realign or change their firing location (in agreement with *Kanter et al., 2017*). This property breaks the symmetry between the maps, as there is still a signature of the original map expressed by grid cells. One implication of this prediction is that remapping is likely to occur to locations in which the synaptic input to the place cells is elevated before the depolarization (even if it is not sufficiently strong to induce firing). Another manifestation of this prediction can be easily seen in *Figure 8—figure supplement 1a*: a margin of positions near the diagonal into which place cells are highly unlikely to remap.

The model explains why artificial remapping occurred under depolarization, but not under hyperpolarization of MEC cells (*Kanter et al., 2017*). In another experiment (*Miao et al., 2015*), remapping in the hippocampus was observed under suppression of activity in the MEC. The reason for the different outcomes (*Kanter et al., 2017*; *Miao et al., 2015*) is unclear. The different outcomes may have arisen from specificity differences in the targeted populations: in *Kanter et al., 2017* chemogenetic receptors were targeted almost exclusively to layer II stellate cells of the MEC in transgenic animals. In comparison, a broader population has been likely affected in *Miao et al., 2015*–in terms of the targeted anatomical area and cortical layers (since the manipulation relied on the diffusion of a virus), and the affected cell types, which most likely included interneurons. Another difference between the experiments of Miao et al and Kanter et al is that the former recorded neural activity in CA3, whereas the latter recorded activity in CA1.

### Experimental predictions

We next discuss some of the predictions arising from our model. To test some of these predictions experimentally, it will be necessary to simultaneously record the activity of multiple cells in the MEC and the hippocampus in freely behaving animals, in sufficient numbers that will enable dynamic decoding of the population activity. Thus, our results offer a rich set of predictions that could be tested using high-throughput recording techniques (*Jun et al., 2017*; *Pfeiffer and Foster, 2015*; *Ziv et al., 2013*; *Zong et al., 2017*).

## Idiothetic path integration

We predict that when updates of the brain's self-estimate of position rely solely on idiothetic path integration, the hippocampal representation of position will lag behind the MEC representation, due to synaptic transmission delays (*Figure 4d*). If verified, this feature of the neural population dynamics may help identify where different types of sensory inputs are integrated into the joint neural representation of position in the hippocampus and the MEC. If sensory inputs that convey spatial information project mainly to hippocampus, we expect a reversal of the time lag under conditions in which such sensory inputs are highly informative.

The hippocampal network in our model is set up such that it expresses bump activity patterns at the neural population level even without inputs from the MEC. This is in agreement with the observation that place fields are expressed under MEC lesions (*Brun et al., 2008*; *Hales et al., 2014*). We predict that under these manipulations, and in conditions in which sensory inputs are absent, such that the brain must rely on idiothetic path integration to update its internal representation of position, bump activity patterns will still be expressed in the hippocampus at the population level, but their association with position will be disrupted. Thus, the spatial fields of single cells will be severely disrupted in these conditions.

## Coupling of entorhinal modules

To test whether a coupling mechanism coordinates the states of grid cell modules, we propose to simultaneously decode activities of multiple modules under conditions in which sensory inputs are absent or poor, and observe whether drifts in these modules are coordinated. A hippocampus-independent mechanism for coordinating the activity of grid cells modules, which relies on synaptic connectivity within the MEC was recently proposed (*Mosheiff and Burak, 2019*). Therefore, it will be of interest to further test whether the coordination requires place cell inputs to the MEC, by repeating the experiment while inactivating inputs from the hippocampus. Under these conditions, cells within a single grid cell module maintain their phase relationships (*Almog et al., 2019*), but it is unknown whether drifts in different modules remain coordinated. Another possible way to approach this question is to examine the coordination of grid cell modules immediately after exposure to a completely novel environment, since it is unlikely that reciprocal connectivity that could support this coordination is established immediately.

## Variability of individual grid cell firing rates across firing fields

Two experiments can directly test the hypothesis that inputs from place cells, independent of sensory inputs, are responsible for a significant fraction of the variability observed in the firing rates of individual grid cells across different firing fields. First, if grid cell modules remain coordinated under hippocampal inactivation, it may be possible to decode position from activity in the MEC in this condition, and test whether firing rates become less variable across firing fields. Second, the independence of variability on sensory inputs could be tested by monitoring activity in the MEC under conditions in which sensory cues are poor or absent. It would also be interesting in this context to measure the variability in animals that have been exposed to different numbers of environments.

## Artificial remapping

We predict that artificial remapping (*Kanter et al., 2017*) arises due to the auto-associative features of network dynamics in the hippocampus, which are believed to arise in area CA3. However, Kanter et al recorded in area CA1. Global and partial remapping have been observed both in CA1 and CA3, yet patterns of activity in CA1 tend to be more correlated across different environments than those observed in CA3 (*Leutgeb et al., 2004*). Thus, partial remapping in CA1 does not necessarily

imply that remapping in CA3 is partial as well. Since Kanter et al observed partial overlap between CA1 activity patterns that were expressed before and after entorhinal depolarization, it will be interesting to record activity also in CA3 under similar conditions. We predict that mixed states will be observed already in CA3 during entorhinal depolarization. Thus, we expect that in CA3 (in similarity to CA1) only a subset of place fields will shift their firing fields. On the other hand, this form of remapping is not expected to occur in the dentate gyrus, since this area is upstream of CA3.

According to our theory, artificial remapping (*Kanter et al., 2017*) arises from expression of population activity patterns that mix contributions normally associated with distinct environments. To test this prediction, we propose to raise animals while exposing them to a limited number of environments. After measuring the response patterns of place cells in each of these environments, we propose to apply depolarization of the MEC in one environment, and look for overlap between place cell activity patterns, and the activity patterns that were previously observed in the other environments (with appropriate controls such as those obtained by randomly permuting cells, or by exposing the animal to new environments after the depolarization experiments). Overlap with one of the experimentally characterized spatial maps is not certain, but this is a plausible outcome of our theory, since we observe that the overlap is highly significant with many spatial maps. If successful, such experiments may provide direct evidence for a key theoretical prediction on the dynamics of auto-associative networks: the emergence of mixed states (*Amit et al., 1985*; *Brunel, 2003*).

## Methods

### Place cell network

The place network consists of $N = 4800$ neurons, which represent positions in $L$ environments. All environments are one-dimensional, spanning a length of 192 cm with periodic boundary conditions. A distinct spatial map is generated for each environment, by choosing a random permutation that assigns all place cells to a set of preferred firing location that uniformly tile this environment.

The synaptic connectivity between place cells is expressed as a sum over contributions from all spatial maps:

$$\mathbf{J} = \sum_{l=1}^{L} J^l \tag{1}$$

where $J_{i,j}^l$ depends on the distance between the preferred firing locations of cells $i$ and $j$ in environment $l$, as follows

$$J_{i,j}^l = \begin{cases} A \exp\left[ -\dfrac{\left(d_{i,j}^l\right)^2}{2\sigma^2} \right] + h & ,i \neq j \\ 0 & ,i = j \end{cases} \tag{2}$$

The first term is an excitatory contribution to the synaptic connectivity that decays with the distance $d_{i,j}^l$, with a Gaussian profile. The second term is a uniform inhibitory contribution. The parameters $A > 0$, $\sigma$, and $h < 0$ are listed in *Table 1*. Note that the connectivity matrices corresponding to any two maps $l$ and $k$ are related to each other by a random permutation:

$$J_{i,j}^l = J_{\pi^{l,k}(i),\pi^{l,k}(j)}^k \tag{3}$$

where $\pi^{l,k}$ denotes the random permutation from map $k$ to map $l$.

### Grid cell network

Each grid cell module is modeled as a double ring attractor (*Xie et al., 2002*) with $n = 960$ neurons. Within each module, neurons are assigned angles $\theta_i$, uniformly distributed in $[0, 2\pi)$ $(i = 1...n)$. The synaptic weight between neurons $i$ and $j$ is given by

$$\mathbf{W}_{i,j} = \begin{cases} B\exp\left[-\dfrac{\left(\left|\theta_i-\theta_j\right|_{2\pi}\pm\Delta\theta\right)^2}{2\rho^2}\right]+k & ,i\neq j \\ \\ 0 & ,i=j \end{cases} \tag{4}$$

where $\left|\theta_i-\theta_j\right|_{2\pi}\equiv\min\left(\left|\theta_i-\theta_j\right|, 2\pi-\left|\theta_i-\theta_j\right|\right)$ and the small phase shift $+\Delta\theta(-\Delta\theta)$ applies to the pre-synaptic neurons with even (odd) index $j$. Thus, evenly indexed and oddly indexed neurons comprise two sub-populations that drive motion of the activity to the right and to the left, respectively, to implement idiothetic path integration (see also *Equation 9* below). The parameters $B>0$, $\rho$, $k<0$, and $\Delta\theta$ are listed in *Table 1*. The ring attractor possesses a continuous manifold of steady states, structured as activity bumps that can be localized anywhere in the range $[0,2\pi]$. We refer to the the center of the activity bump as the phase of the population activity pattern. Grid cells from distinct modules are not directly connected to each other.

In our implementation of the model, we include three grid cell modules with grid spacings $\lambda^\mu = 64$, 48 and 38.4 cm for modules $\mu = 1$, 2 and 3, respectively. Within each environment, positions are mapped to attractor states by tiling the possible phases $[0,2\pi]$ of the population activity periodically along the extent of the environment. For simplicity, we chose grid spacings that correspond precisely to 5, 4 and 3 cycles along the 192 cm environment. To account for grid cell realignment during global remapping (*Fyhn et al., 2007*), a uniform phase shift $\Delta^{l,\mu}$ is applied to the tiling of grid phases to positions. This phase shift is randomly chosen from the range $[0,2\pi)$, independently for each environment $l$ and module $\mu$.

## Bi-directional synaptic connectivity between grid cells and place cells

The mutual connection between each place cell and grid cell depends linearly on the overlap between their tuning curves, summed over all environments. We thus define a connectivity matrix $\mathbf{M}^\mu$ between all place cells and all grid cells from module $\mu$ as a sum over contributions $M^{l,\mu}$ from all environments:

$$\mathbf{M}^\mu = \sum_{l=1}^{L} M^{l,\mu} \tag{5}$$

To determine $M^{l,\mu}$ we first define a correlation matrix

$$\mathbf{m}_{i,j}^{l,\mu} = \frac{1}{z^\mu}\int f_i^l(x)\cdot g_j^{l,\mu}(x)\mathrm{d}x \tag{6}$$

where $f_i^l$ is the idealized receptive field of place cell $i$ and $g_j^{l,\mu}$ is the idealized receptive field of grid cell $j$ measured from uncoupled networks at environment $l$. The normalization factor $z^\mu$ is chosen such that $\mathbf{m}^{l,\mu}\in[0,1]\,\forall\mu$.

Note that the correlation matrices corresponding to any two maps $l$ and $k$ are related to each other by a permutation on the place cell indices, and a cyclic shift on the grid cell indices. Finally,

$$M^{l,\mu} = \alpha\cdot\mathbf{m}^{l,\mu}+\beta \tag{7}$$

where the parameters $\alpha>0$ and $\beta<0$ are identical for all modules and maps (*Table 1*).

The idealized receptive fields of place cells and grid cells from *Equation (6)* are determined as follows: the field $f_i^l$ is generated by simulating only the place cell network, uncoupled from the grid cell network and when only a single map is embedded in the connectivity. This generates an idealized bump around the initial condition, and all possible steady states of this bump are related to each other by rigid translation in the neural space sorted by the coordinates of map $l$. Thus, rigid translations of the observed activity bump are associated with all possible positions in the environment. Similarly, the tuning curve $g_j^{l,\mu}$ of a particular grid cell from module $\mu$, is defined using a rigid translation of an idealized grid cell activity pattern which emerged in a simulation of an uncoupled grid cell module network, and association of the different translations with all possible positions in the environment.

## Dynamics

The dynamics of neural activity are described by a standard rate model. The synaptic activation $S_i$ of place cell $i$ evolves in time according to the following equation:

$$\tau \dot{S}_i = -S_i + \phi\left[\sum_{j=1}^{N}\mathbf{J}_{i,j}S_j + \gamma_g\sum_{\mu}\sum_{k=1}^{n}\mathbf{M}_{i,k}^{\mu}s_k^{\mu} + I_{\text{pc}}\right] + \eta_i \tag{8}$$

and the synaptic activation $s_i^{\mu}$ of grid cell $i$ from module $\mu$ evolve as follows:

$$\tau \dot{s}_i^{\mu} = -s_i^{\mu} + \phi\left[\sum_{j=1}^{n}\mathbf{W}_{i,j}s_j^{\mu} + \gamma_p\sum_{k=1}^{N}\mathbf{M}_{i,k}^{\mu^{\mathrm{T}}}S_k + I_{\text{gc}}^{\mu} \pm \varepsilon^{\mu}\cdot v\right] + \eta_i^{\mu} \tag{9}$$

In both equations, $\tau$ is the synaptic time constant (taken for simplicity to be identical for all synapses). The coupling parameters $\gamma_g$ and $\gamma_p$ determine the strength of synaptic connections from grid cells to place cells and from place cells to grid cells, respectively. The velocity signal is denoted by $v$. The signs $(+)$ and $(-)$ are used for the even and odd grid cell populations, respectively. Thus, neurons with even index, whose outgoing synaptic weights are biased to the right (see *Equation 4*) are preferentially activated during motion to the right, whereas neurons with odd index, whose synaptic weights are biased to the left are preferentially activated during motion to the left. The coefficients $\varepsilon^{\mu}$ determine the weighting of the velocity signal per module. These parameters are chosen in order to obtain correct grid spacings during idiothetic path integration. The external currents $I_{\text{pc}}$ in the place cell population and $I_{\text{gc}}^{\mu}$ in the grid cell population are constant in time and identical in all cells from a given subpopulation. These currents include two terms: one term is the baseline current, required to drive activity when a single spatial map is embedded in the connectivity. The second term compensates for interference coming from additional maps, and is proportional to $L-1$ (see Appendix 1, *Equations 37 and 38*).

The transfer function $\phi$ determines the firing rate (in Hz) of place cells $(R_i)$ and grid cells $(r_i^{\mu})$ as a function of their total synaptic inputs. To resemble realistic neuronal F-I curves, it is chosen to be sub-linear:

$$\phi(x) = \begin{cases} 0 & ,x \leq 0 \\ \sqrt{x} & ,x > 0 \end{cases} \tag{10}$$

The noise terms $\eta_i$ in *Equation 8* and $\eta_i^{\mu}$ in *Equation 9* are included only in *Figure 5*. These terms have zero mean, and to mimic Poisson noise, we assume that they are independent for different neurons and obey

$$\langle \eta_i(t)\eta_i(t')\rangle = R_i(t)\delta(t-t') \tag{11}$$

$$\langle \eta_i^{\mu}(t)\eta_i^{\mu}(t')\rangle = r_i^{\mu}(t)\delta(t-t') \tag{12}$$

We implemented the dynamics using the Euler-method for numeric integration, with a time interval $\Delta t$ (*Table 1*).

## Coupling parameters

The parameter $\alpha$ in *Equation 7* is included for convenience but is redundant since it can be absorbed in the definitions of $\beta$, $\gamma_g$ and $\gamma_p$. We note also that due to the coupling parameters $\gamma_g$ and $\gamma_p$ the synaptic connectivity is not necessarily symmetric. However, it is possible to recast the equations in a form that involves symmetric connectivity, by rescaling the synaptic variables $S_i$ and $s_i^{\mu}$. This observation is important since it implies that the dynamics always settle on stationary steady states and do not exhibit limit cycles (*Cohen and Grossberg, 1983*).

The coupling parameters $\gamma_g$ and $\gamma_p$ were chosen based on the following considerations: on one hand, if the coupling parameters are too weak, the network exhibits persistent states in which the bump positions in different sub-networks are incompatible. Moreover, it is essential for path integration and the coupling of grid cell modules, that both place cells can influence the position of grid cell bumps, and that grid cells can influence the state of the place cell network. Thus, it was not

sufficient to include connections only in one direction. On the other hand, the connections cannot be too strong since we are interested in the regime in which the structure of steady states in each sub-network is predominantly determined by its internal connectivity. These goals can be achieved using a wide range of choices for $\gamma_g$ and $\gamma_p$. A single set of parameters (*Table 1*) was used throughout the manuscript.

## Bump score and location analysis

Without loss of generality, whenever we set the system in a bump state, we do so in the map labeled as map 1.

### Place cell network

To identify whether the place cell network expresses a bump state, and to identify its location $x$ and associated spatial map $l$, we define a correlation coefficient $q^l(x)$ that quantifies the overlap between the hippocampal population activity pattern and the activity pattern corresponding to position $x$ in spatial map $l$:

$$q^l(x) = \sum_i P_i^l(x) \cdot R_i \qquad (13)$$

where $R_i$ (already defined above) is the firing rate of place cell $i$, and $P_i^l(x)$, is the firing rate of neuron $i$ in an idealized bump state localized at position $x$ in map $l$. The idealized bump (as defined above) is obtained from the activity of a network in which a single map (map $l$) is embedded in the neural connectivity, and there is no quenched noise.

Next, we define a bump score for each spatial map, defined as the maximum of the correlation coefficient $q^l(x)$ over all positions $x$ in spatial map $l$:

$$Q^l = \max_x q^l(x) \qquad (14)$$

Finally, the map with the highest $Q^l$ value is considered as the winning map, and the location $x$ that generated that value is considered as the location of the place cell bump within that map.

### Grid cell network

To identify the location corresponding to activity in each grid cell module, we use a similar procedure, in which we calculate the overlap between the population activity and an idealized activity pattern of grid cells, evaluated across all positions in the winning map. Note that due the periodicity of grid cell responses, multiple positions produce the same correlation coefficient. When measuring the distance between the grid cell activity bump and the place activity bump or with its initial position (for example in *Figure 3c,g and k*), we chose from these periodically spaced positions the one closest to the location of the place cell bump.

## Grid cells hyperpolarization and depolarization implementation

To mimic to effects of transgenic DREADDs (Designer Receptors Exclusively Activated by Designer Drugs) in MEC layer 2 cells as performed experimentally (*Kanter et al., 2017*) we added a constant current $I_{\mathrm{per}}$ to the total synaptic input driving grid cell activity. *Equation 9* is thus replaced by:

$$\tau \dot{s}_i^\mu = -s_i^\mu + \phi\left[\sum_{j=1}^n \mathbf{W}_{i,j} s_j^\mu + \gamma_p \sum_{k=1}^N \mathbf{M}_{i,k}^{\mu^{\mathrm{T}}} S_k + I_{\mathrm{gc}}^\mu \pm \varepsilon^\mu \cdot v + I_{\mathrm{per}}\right] + \eta_i^\mu \qquad (15)$$

A current $I_{\mathrm{per}} > 0$ was used for depolarization, and a current $I_{\mathrm{per}} < 0$ was used for hyperpolarization (see *Table 1*).

### Analysis of persistent mixed states

In *Figure 7*, the system is placed at bump states corresponding to 500 uniformly distributed spatial locations in map 1, and its state is analyzed after a 2 s delay period following the onset of grid cell depolarization.

## Classification of place cell responses under grid cell depolarization

Classification of place cell responses under grid cell depolarization was performed based on the following criteria: (a) Cells that turned off (~6%) are defined as cells that exhibited a precisely vanishing firing rate at all locations. (b) Cells with a minor field (~1.5%) are defined as cells that exhibited maximal peak firing rate of 2.5 Hz (not necessarily in the vicinity of the peak's original location). (c) Cells that rate remapped (~7%) are defined as cells whose peak firing rate remained within a distance of at most 12 cm from its original location. In addition, their peak firing rate decreased or increased by more than 33% relative to its baseline value (but did not decrease below 2.5 Hz). (d) Cells that were not affected (~11%) are defined as in (c), except that their peak firing rate did not change by more than 33% relative to its baseline value. (e) Cells that shifted their firing location (~29%) are defined as cells that exhibited a peak firing rate greater than 2.5 Hz, at a distance of 12 cm or more from the baseline peak location. (f) Cells that expressed remapped fields but maintained their firing field in their original location (~36.5%) are defined as in (e), but exhibited in addition a peak firing rate which did not change by more than 33% relative to its baseline value and at a distance not greater than 12 cm from its original location. (g) Cells that expressed multiple fields (~9%) are defined as cells that exhibited more than a single peak (at least one peak greater than 2.5 Hz - could involve in addition multiple smaller peaks) at distances greater than 12 cm from the original peak location.

## Acknowledgements

This research was supported by the Israel Science Foundation grant No. 1745/18, in part by the Israel Science Foundation grant No. 1978/13, and by a grant from the GIF, the German-Israeli Foundation for Scientific Research and Development. We acknowledge support from the Gatsby Charitable Foundation.

## Additional information

### Funding

| Funder | Grant reference number | Author |
| --- | --- | --- |
| Israel Science Foundation | 1745/18 | Haggai Agmon Yoram Burak |
| German-Israeli Foundation for Scientific Research and Development | I-1477-421.13/2018 | Haggai Agmon Yoram Burak |
| Gatsby Charitable Foundation | Gatsby Program in Theoretical Neuroscience at the Hebrew University | Haggai Agmon Yoram Burak |
| Israel Science Foundation | 1978/13 | Haggai Agmon Yoram Burak |

The funders had no role in study design, data collection and interpretation, or the decision to submit the work for publication.

### Author contributions

Haggai Agmon, Conceptualization, Software, Formal analysis, Validation, Investigation, Methodology, Writing - original draft, Project administration, Writing - review and editing, Development of the theory; Yoram Burak, Conceptualization, Supervision, Funding acquisition, Investigation, Methodology, Writing - original draft, Project administration, Writing - review and editing, Development of the theory

### Author ORCIDs

Haggai Agmon https://orcid.org/0000-0002-7212-9052
Yoram Burak https://orcid.org/0000-0003-1198-8782

**Decision letter and Author response**
Decision letter https://doi.org/10.7554/eLife.56894.sa1
Author response https://doi.org/10.7554/eLife.56894.sa2

## Additional files

### Supplementary files
• Transparent reporting form

### Data availability

This is a theoretical manuscript which does not contain data of our own. The rat trajectory used in Figure 4 to generate a distribution of velocities is taken from Fig.2c in (Hafting et al., 2005). It is available online at https://doi.org/10.11582/2014.00001.

The following previously published dataset was used:

| Author(s) | Year | Dataset title | Dataset URL | Database and Identifier |
|---|---|---|---|---|
| T Hafting, M Fyhn, S Molden, MB Moser, EI Moser | 2005 | Grid cell data Hafting et al 2005 | https://doi.org/10.11582/2014.00001 | NIRD Research Data Archive, 10.11582/2014.00001 |

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

# Appendix 1

In this appendix, we analyze mathematically how the embedding of multiple spatial maps in the synaptic connectivity affects the existence of bump states in the system, using a mean-field analysis. Similar questions have been extensively studied for associative networks with discrete memories, and for models of multiple spatial map representations in the hippocampus. Here, we consider properties of the coupled entorhinal-hippocampal network.

As the number of spatial maps $L$ increases, the shape of the bump is increasingly distorted until at a certain critical $L_c$ the representation of the bump will totally collapse and no localized bump will be observed in any of the embedded maps. This happens since with every map added, there is addition of quenched noise into the place cell connections, arising from the random permutations and thus, there is accumulating interference between the distinct map representations. To characterize the influence of the quenched noise, we compute here the mean and variance of quenched noisy inputs to place cells and grid cells.

The calculation of the mean input is used to adjust the feed-forward inhibition in the network (which is uniform in each grid cell module and within the place cell network), such that the mean input to all cells remains fixed with addition of new maps (*Equations 37 and 38*). The calculation of the variance allows us to assess how $L_c$ scales with network parameters.

## Quenched noise conveyed to place cells

Without loss of generality, we assume a steady bump state in map 1. Since the synaptic activities are steady, we can replace them in all expressions with the corresponding firing rates: $S_i = R_i$ and $s_i = r_i$ . The firing rate profile of place cells can be thought as the sum of two contributions: a smooth bump of synaptic activation associated with map 1, and noisy deviations originating from the synaptic weights associated with the rest of the maps. When $L \geq L_c$, these deviations are too large and the representation collapses.

Since we assume that the deviations are sufficiently small, we decompose terms in the steady state equation to inputs from map 1 and to inputs from all the other interfering maps. We obtain terms describing (1) inputs from place cells through the synaptic connectivity that corresponds to map 1, (2) noisy inputs from place cells arising from all the other interfering maps, (3) inputs from grid cells, associated with map 1, and (4) noisy inputs from grid cells that arise from all the other interfering maps. When evaluating these terms, to leading order in the deviations we can approximate the firing rates $R_i, r_i$ by those taken from the case $L = 1$. Rewriting the steady state equation of a place cell accordingly yields

$$R_i = \phi\left[\underbrace{\sum_{j=1}^{N} J_{i,j}^1 R_j}_{1} + \underbrace{\sum_{l=2}^{L}\sum_{k=1}^{N} J_{i,k}^l R_k}_{2} + \gamma_g \sum_{\mu}\left(\underbrace{\sum_{h=1}^{n} M_{i,h}^{1,\mu} r_h^{\mu}}_{3} + \underbrace{\sum_{m=2}^{L}\sum_{f=1}^{n} M_{i,f}^{m,\mu} r_f^{\mu}}_{4}\right) + I_{\text{pc}}\right] + \eta_i \qquad (16)$$

We are interested in the means, variances and covariances of terms (2) and (4), compared to the means of terms (1) and (3).

## Quenched noise conveyed to place cells by place cells (term 2):

We note that the sum of each row and the sum of each column of $J^l$ are all identical, and denote

$$\sum_{j} J_{i,j}^l = \sum_{i} J_{i,j}^l \equiv C \quad ; \quad \forall i, j \qquad (17)$$

Since maps are drawn independently, we note also that

$$\langle J^l J^m \rangle = \langle J^l \rangle \langle J^m \rangle \quad ; \quad \forall l \neq m \qquad (18)$$

where the pointy brackets denote an average over the random permutation $\pi^l$ that relates the matrix $J^l$ to $J^1$:

$$J^l = \mathcal{P}_{\pi^l} J^1 \mathcal{P}_{\pi^l}^{\mathrm{T}} \tag{19}$$

and $\mathcal{P}_\pi$ is the permutation matrix corresponding to the permutation $\pi$.

For convenience, we denote term (2) by $P$:

$$P = \sum_{l=2}^{L} \sum_{k=1}^{N} J_{i,k}^l R_k \tag{20}$$

The mean of $P$ is

$$\langle P \rangle = \sum_{l=2}^{L} \sum_{k=1}^{N} \langle J_{i,k}^l \rangle R_k = \sum_{l=2}^{L} \sum_{k=1}^{N} \frac{C}{N} R_k = (L-1) C \bar{R} \tag{21}$$

where $\bar{R}$ is the average firing rate of place cells in the bump steady state when $L = 1$. Since the permutations in different maps are drawn independently, the variance of $P$ is

$$\mathrm{Var}(P) = (L-1) \left( \left\langle \left[ \sum_{k=1}^{N} J_{i,k}^l R_k \right]^2 \right\rangle - C^2 \bar{R}^2 \right) \tag{22}$$

Thus, both the mean and the variance grow linearly with $L$, and vanish when $L = 1$.

## Quenched noise conveyed to place cells by grid cells (term 4):

Similarly, the sums of all rows of $\mathbf{m}^{l,\mu}$ (*Equation 6*) are equal to each other, and we denote

$$\sum_{i=1}^{n} \mathbf{m}_{i,j}^{l,\mu} \equiv \tilde{D}^\mu \quad ; \quad \forall j \tag{23}$$

Thus,

$$\sum_{i=1}^{n} M_{i,f}^{l,\mu} = \alpha \tilde{D}^\mu + \beta n \equiv D^\mu \quad ; \quad \forall f \tag{24}$$

Similarly to (*Equation 18*), note that

$$\langle M^{l,\mu} M^{m,\mu} \rangle = \langle M^{l,\mu} \rangle \langle M^{m,\mu} \rangle \quad ; \quad \forall l \neq m \tag{25}$$

For convenience, we denote the contributions of module $\mu$ from term 4 by $G^\mu$:

$$G^\mu = \sum_{m=2}^{L} \sum_{f=1}^{n} M_{i,f}^{m,\mu} r_f^\mu \tag{26}$$

The mean contribution of module $\mu$ to the quenched noise in the input to place cells is

$$\gamma_g \langle G^\mu \rangle_{\mathrm{per}} = \gamma_g \sum_{m=2}^{L} \sum_{f=1}^{n} \langle M_{i,f}^{m,\mu} \rangle_{\mathrm{per}} r_f^\mu = \gamma_g (L-1) D^\mu \bar{r}^\mu \tag{27}$$

The variance of place cell inputs arising from grid cell activities includes two types of contributions: (a) covariance terms between inputs arising from all pairs of grid modules, and (b) covariance terms between place cell inputs and the inputs from each grid module (discussed in the next subsection).

Since phase shifts of each grid module are drawn independently of each other we note that $\langle M^{l,\mu} M^{l,\nu} \rangle = \langle M^{l,\mu} \rangle \langle M^{l,\nu} \rangle$ for $\mu \neq \nu$. Consequently, all covariance terms between distinct modules (the mixed expressions in term 4) vanish (*Appendix 1—figure 1a*). The variance to place cell inputs from each grid module is:

$$\gamma_g^2 \text{Var}(G^\mu) = (L-1)\gamma_g^2 \left( \left\langle \left[ \sum_{f=1}^n M_{i,f}^{l,\mu} r_f^\mu \right]^2 \right\rangle_{\text{per}} - (D^\mu)^2 (\bar{r}^\mu)^2 \right) \tag{28}$$

where $\bar{r}^\mu$ is the average firing rate of grid cells from module $\mu$ in the bump steady state when $L = 1$.

## Overall mean and variance of place cell inputs

The overall mean of the quenched noise is (sum of terms from *Equation (21)* and *Equation(27)*):

$$(L-1) \cdot \left[ C\bar{R} + \gamma_g \sum_\mu D^\mu \bar{r}^\mu \right] \tag{29}$$

The overall variance of the quenched noise is determined by *Equation (22)*, *Equation (28)* and the covariance between place cell and grid cell activities (terms 2 and 4 in *Equation 16*). Since in each map, phase shifts of each grid module are drawn independently from the permutation of the place cells, we note that $\langle J^l M^{l,\mu} \rangle = \langle J^l \rangle \langle M^{l,\mu} \rangle$, and thus all covariance terms between place cell and grid cell vanish (*Appendix 1—figure 1b*). The overall variance is thus:

$$\text{Var}(P) + \gamma_g^2 \sum_\mu \text{Var}(G^\mu) \tag{30}$$

Note that all terms in these expressions are proportional to $(L-1)$.

## Quenched noise conveyed to grid cells

Similarly to our derivation for place cells, we decompose the input to grid cells into a contribution from map 1 (in which we assume a bump state), and to inputs from all other interfering maps:

$$r_i^\mu = \phi \left[ \sum_{j=1}^n W_{i,j} r_j^\mu + \gamma_p \left( \sum_{k=1}^N M_{i,k}^{1,\mu^T} R_k + \sum_{l=2}^L \sum_{h=1}^N M_{i,h}^{l,\mu^T} R_h \right) + I_{\text{gc}}^\mu \right] + \eta_i^\mu \tag{31}$$

The quenched noise inputs to grid cells originate solely from place cells. Thus, we are interested in the mean and variance of the third term, compared to the sum of the first two terms in the argument of $\phi$. We denote

$$g^\mu = \sum_{l=2}^L \sum_{h=1}^N M_{i,h}^{l,\mu^T} R_h \tag{32}$$

We note that

$$\sum_{i=1}^N m_{i,j}^{l,\mu^T} \equiv \tilde{E}^\mu \quad ; \quad \forall j \tag{33}$$

and thus

$$\sum_{i=1}^N M_{i,f}^{l,\mu^T} = \alpha \tilde{E}^\mu + \beta N \equiv E^\mu \quad ; \quad \forall f \tag{34}$$

From here on, the derivation is similar to the previous calculation and it is straightforward to see that the mean of the quenched noise input to grid cell is equal to

$$\gamma_p \langle g^\mu \rangle_{\text{per}} = \gamma_p (L-1) E^\mu \bar{R} \tag{35}$$

and the variance is

$$\gamma_p^2 \text{Var}(g^\mu) = \gamma_p^2 (L-1) \left( \left\langle \left[ \sum_{h=1}^N M_{i,h}^{l,\mu^T} R_h \right]^2 \right\rangle_{\text{per}} - (E^\mu)^2 \bar{R}^2 \right) \tag{36}$$

## Compensating for the mean of the quenched noise

In order to sustain a constant mean input to place cells and grid cells as $L$ is varied, we adjust the external current to compensate for the mean of the quenched noise, *Equation 29* and *Equation 35*. Thus, the external input to place cells is

$$I_{\mathrm{pc}}(L) = I_{\mathrm{pc},0} - (L-1)\left[C\bar{R} + \gamma_g \sum_{\mu} D^{\mu}\bar{r}^{\mu}\right] \tag{37}$$

and the external input to each grid cell module $\mu$ is

$$I_{\mathrm{gc}}^{\mu}(L) = I_{\mathrm{gc},0}^{\mu} - (L-1)\gamma_p E^{\mu}\bar{R} \tag{38}$$

where $I_{\mathrm{pc},0}$ and $I_{\mathrm{gc},0}^{\mu}$ are independent of $L$.

## Scaling of the variance of inputs with $N, n$, and assessing $L_c$

We now consider how the variance of inputs scales with the number of neurons in the system, when the fraction of active cells and their typical firing rates are kept fixed with addition of new spatial maps.

To maintain a fixed distribution of place cell and grid cell firing rates at steady state under the assumption that $N \gg 1$ place cells and $n \gg 1$ grid cells in each module cover uniformly and densely the environment, the synaptic connectivity should scale as follows under re-scaling of $N$ and $n$:

$$J_{i,j} \propto \frac{1}{N}, \ \ W_{i,j} \propto \frac{1}{n}, \ \ \gamma_g \propto \frac{1}{n}, \ \ \gamma_p \propto \frac{1}{N} \tag{39}$$

We focus first on the inputs to place cells: the signal conveyed to place cells in *Equation (16)*, which we denote by $\mathcal{S}$, is composed of the contribution from terms 1 and 3. The noise conveyed to place cells, denoted by $\mathcal{N}$, is composed of the contribution from terms 2 and 4. With the scaling of *Equation 39*, $\mathcal{S}$ is independent of the number of neurons when $N$ and $n$ are large.

The mean of the noise $\mathcal{N}$ is canceled by the adjustment of the external current (*Equation 37*),

$$\langle \mathcal{N} \rangle_{\mathrm{per}} = 0 \tag{40}$$

and the variance of the noise is given in *Equation (30)*:

$$\mathrm{Var}(\mathcal{N}) = \mathrm{Var}(P) + \gamma_g^2 \sum_{\mu} \mathrm{Var}(G^{\mu}) \tag{41}$$

In all these terms, the variance is across random choices of the place cell permutations and grid cell phases. Due to the random shuffling of place cells under random permutations, it is straightforward to show (see also *Appendix 1—figure 2a*) that

$$\mathrm{Var}(P) \propto \frac{(L-1)}{N} \tag{42}$$

Therefore, in a system composed only of place cells, $L_c \propto N$.

However, the variance of place cell inputs arising from grid cell activities ($\mathrm{Var}(G^{\mu})$) is constant (for $n \gg 1$) and does not depend on the the number of neurons (*Appendix 1—figure 2b*). This is due to the fixed phases between pairs of grid cells that belong to the same module in each of the maps. Therefore, increasing the number of grid cells would not increase $L_c$. Overall, as the total number of neurons increases, $L_c$ increases until the variance of place cell inputs is dominated by the variance arising from grid cell activities.

As for the grid cells: similarly to place cells, the signal (first two terms in *Equation 31*) is independent of the number of neurons when $N$ and $n$ are large, and the mean of the noise (third term in *Equation 31*) is canceled by the adjustment of the external current (*Equation 38*). Similarly to place cells, the variance of the noise of each grid cell module input (*Equation 36*), which arise from place cell activities, is inversely proportional to the number of place cells (see also *Appendix 1—figure 3*):

$$\mathrm{Var}(g^{\mu}) \propto \frac{(L-1)}{N} \tag{43}$$

To qualitatively assess $L_c$ numerically, *Appendix 1—figure 4* shows the bump score as a function of $L$ when initial rates were set as bumps (consistent condition) in map 1 (without loss of generality), for the fixed number of neurons used throughout this manuscript and for three different values of $\gamma_g$. The bump score decreases sharply at a certain characteristic value of $L$, which provides an assessment of $L_c$. As expected and based on (*Equation 41*), the capacity increases when decreasing $\gamma_g$ (the coupling parameter that determines the strength of connectivity from grid cells to place cells).

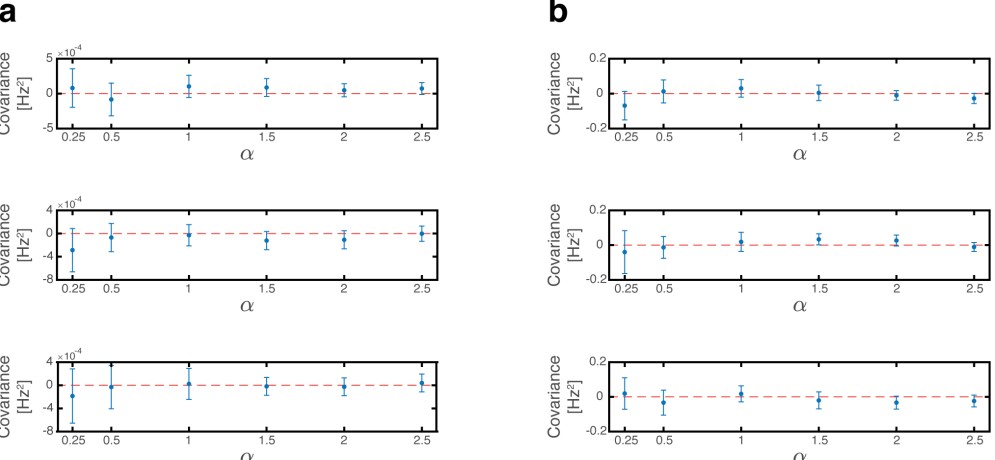

**Appendix 1—figure 1.** Covariances of place cell inputs across sub-populations vanish. (**a**) Simulation results (blue dots) showing the covariance of place cell inputs arising from activities of pairs of different grid cell modules. A common scaling factor $\alpha$ determines the number of place cells and grid cells in each module where $\alpha = 1$ was used throughout this manuscript. Red dashed lines show the zero-identity line for reference. Top: covariance between first and second modules. Middle: covariance between first and third modules. Bottom: covariance between second and third modules. Simulation results include 2000 random map realizations bootstrapped into 40 batches of 50 realizations to obtain multiple covariance measurements. Error bars are 1.96 times the standard deviations obtained from each simulation, divided by the square root of the number of realizations (corresponding to a confidence interval of 95%). (**b**) Same as (**a**) but for the covariance of place cell inputs arising from activities of place cells and each of the grid cell modules. Top: covariance between place cells and first grid cell module. Middle: covariance between place cells and second grid cell module. Bottom: covariance between place cells and third grid cell module.

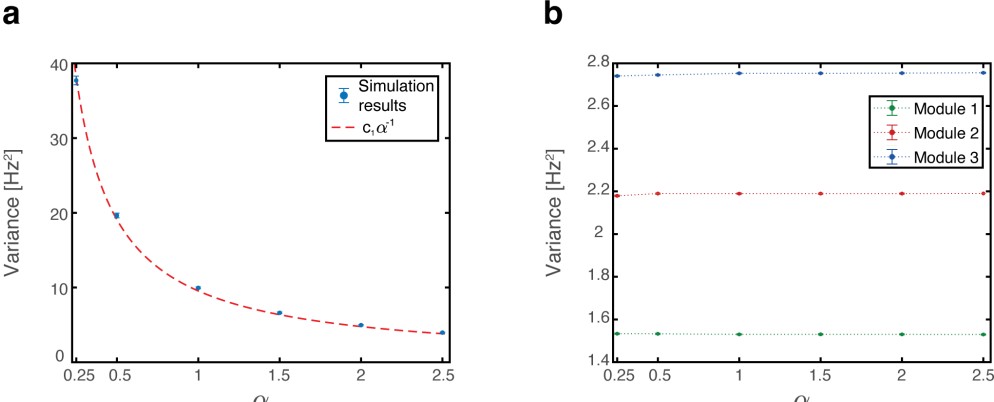

**Appendix 1—figure 2.** Variance of place cell inputs. (**a**) Simulation results (blue dots) showing the variance of place cell inputs arising from place cell activities. A common scaling factor $\alpha$ determines the number of place cells and grid cells in each module where $\alpha = 1$ was used throughout this manuscript. Simulation results include 2000 random map realizations bootstrapped into 40 batches of 50 realizations to obtain multiple variance measurements. Error bars are 1.96 times the standard deviations obtained from each simulation, divided by the square root of the number of realizations

(corresponding to a confidence interval of 95%. Some error bars are too small to be visible). Dashed red line shows the curve $c_1\alpha^{-1}$ with optimal fitted $c_1 = 9.543$ for reference, $R^2 = 0.999$. (**b**) Same as (**a**) but for the variance of place cell inputs arising from grid cell activities.

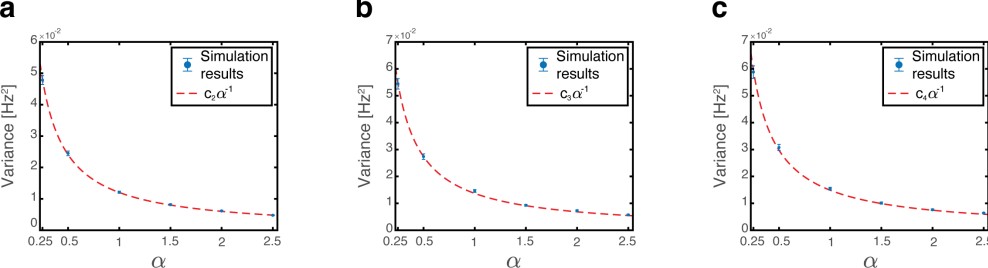

**Appendix 1—figure 3.** Variance of grid cell inputs. (**a**) Simulation results (blue dots) showing the variance of grid module 1 inputs arising from place cell activities. A common scaling factor $\alpha$ determines the number of place cells and grid cells in each module where $\alpha = 1$ was used throughout this manuscript. Simulation results include 2000 random map realizations bootstrapped into 40 batches of 50 realizations to obtain multiple variance measurements. Error bars are 1.96 times the standard deviations obtained from each simulation, divided by the square root of the number of realizations (corresponding to a confidence interval of 95%. Some error bars are too small to be visible). Dashed red line shows the curve $c_2\alpha^{-1}$ with optimal fitted $c_2 = 0.01203$ for reference, $R^2 = 0.999$. (**b**) Same as (**a**) but for grid module 2. Dashed red line shows the curve $c_3\alpha^{-1}$ with optimal fitted $c_3 = 0.01369$ for reference, $R^2 = 0.999$. (**c**) Same as (**a**) but for grid module 3. Dashed red line shows the curve $c_4\alpha^{-1}$ with optimal fitted $c_4 = 0.01488$ for reference, $R^2 = 0.999$.

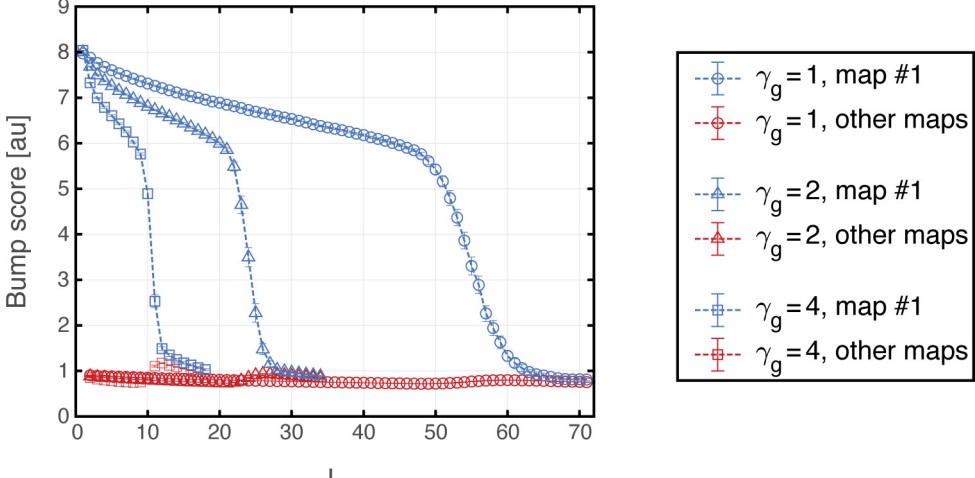

**Appendix 1—figure 4.** Capacity dependence on grid to place coupling parameter. Bump score vs the total number of embedded maps for three different values of $\gamma_g$ (circle, triangle and square) when starting from a 'consistent' initial condition in map #1. Blue curves show the bump score for map #1 and overlapping red curves show the average bump score for the rest of the embedded maps (whenever $L > 1$). Each plotted value is the average bump score obtained from 100 randomly chosen realizations for each specific value of $L$ (and including 5 randomly chosen initial conditions within each realization). Error bars are 1.96 times the standard deviations obtained from each simulation, divided by the square root of the number of realizations (corresponding to a confidence interval of 95%. Some error bars are too small to be visible).

