## [Decision Letter]

**Acceptance summary:**

Grid cells and place cells are populations of neurons that are thought to work together to represent an animal's spatial position using complementary encodings of space. This paper provides a simple but compelling mechanistic model of how they might interact and, in doing so, suggests explanations for a variety of experimental findings.

**Decision letter after peer review:**

Thank you for submitting your article "Bidirectional coupling of hippocampus and MEC accounts for grid field variability and artificial place cell remapping" for consideration by *eLife*. Your article has been reviewed by Laura Colgin as the Senior Editor, a Reviewing Editor, and three reviewers. The reviewers have opted to remain anonymous.

The reviewers have discussed the reviews with one another and the Reviewing Editor has drafted this decision to help you prepare a revised submission.

We would like to draw your attention to changes in our revision policy that we have made in response to COVID-19 (https://elifesciences.org/articles/57162). Specifically, we are asking editors to accept without delay manuscripts, like yours, that they judge can stand as *eLife* papers without additional data, even if they feel that they would make the manuscript stronger. Thus the revisions requested below primarily address clarity and presentation and require only minor additional numerical simulations.

Summary:

Agmon and Burak present a novel integrated model of grid cell and place cell networks, with bidirectional coupling between grid cells in the medial entorhinal cortex and place cells in the hippocampus. The model combines attractor dynamics in the hippocampus and in multiple grid modules in an interesting and consistent manner. Despite its relative simplicity, the joint place cell-grid cell network can explain a wide range of experimental findings, including that hippocampal place cells and hippocampal remapping can persist without MEC, the variability of firing rates across grid fields, and the intriguing observation that artificial hippocampal remapping is induced by depolarization but not hyperpolarization of MEC inputs to the hippocampus.

Overall, the model surpasses previous feedforward models, is implemented at an appropriate level, is presented in language that is accessible to experimentalists and is quite timely given recent puzzling observations on the relation between grid cells and place cells.

Essential revisions:

1) The operation of the network should be presented more thoroughly. The development of this combined network is a major result, but it is a bit unclear how the network produces the reported results. In particular:

a) Illustrating bump dynamics. The only dynamical data presented in the paper relates to path-integration and drift. When a bump develops from random initial conditions, does it do so steadily or in spurts? Similarly, how does bump score across different maps evolve when the bump develops into a mixed state?

b) How many maps can be stored by the network? In the Supplementary Material, the authors calculate that this number scales with the number of neurons. Do simulations support this result, and what is the scaling prefactor?

c) What range of parameters, especially the relative strength of connections within and across subnetworks, produce desired results?

2) How does the place cell network operate without grid cell input, and how does this compare to MEC lesion experiments? What activity patterns appear? I presume that it would not path-integrate-is this true? Also, it seems that place cell remapping arises from grid cell inputs, namely, map-dependent shifts in grid cell phase – is this true?

3) All the modeling presented in the paper is done in 1-d. It would be useful to confirm that it indeed also works in 2-d (and use the grid scales consistent with the experimental data). This would facilitate the comparison to (future) experimental results.

4) In addition to artificial remapping, where the firing rate distribution across grid fields is experimentally manipulated, a reorganization of the firing rate among grid fields can also be naturally observed in rate remapping paradigms (Diehl et al., 2017). Although the authors may have omitted this experimental finding because it does not fit into their framework of the place cell network mimicking global remapping, some of their analysis of artificial remapping shows that their network can in principle generate persistent mixed states. More in depth analysis should be added which describes to what extent the findings by Diehl et al. can be reproduced by the current network architecture or whether major modifications would be necessary. The Introduction, Results and Discussion sections should be expanded to include these analyses and their interpretation.

5) Correction of drifts: It is interesting that the network can correct the drift of the grid module. There has now been a literature on this issue (Hardcastle, Ganguli and Giocomo, 2015; Mulas, Waniek and Conradt, 2016; Pollock et al., 2018). In particular these studies have shown that error correction can happen between the boundary cell or landmark cells to grid cells. It is unclear if/how the current error-correction mechanism fundamentally differs from the previous studies. It would be useful to clarify these issues. Also, how is "incompatible drift" defined in subsection “Coordination of the grid cell representation: suppression of incompatible drifts”?

6) Robustness with respect to the selection of the grid modules: The authors used [38.4,48, 64]cm for the 3 grid modules. Are the predictions robust with respect to the selection of the modules? For example, the form of decoding error using ML estimate would depend on the how the scales of the module are organized. With a hierarchical grid code (Mathis et al., 2012; Wei et al., *eLife* 2015), it is much less prone to non-local errors, unlike the residual number system studied by Fiete et al., (2008).

7) One highlight of the current work is that the model could account for the seemingly paradoxical finding in Kanter et al., (2017). However, as the authors pointed out that there is also the issue regarding the generality of the finding in Kanter et al., (2017). Is there a regime that the proposed model can also account for Miao et al., (2015)? The authors wrote: "while in Ref (Kanter et al., 2017) chemogenetic receptors were targeted almost exclusively to layer II stellate cells of the MEC in transgenic animals, a broader population has been likely affected using viral injection in Ref (Miao et al., 2015)." The argument may not address the problem, because both stellate cells and pyramidal cells in EC2 can be grid cells. These points need to be more carefully discussed, as it is critical to the test of the proposed model.

8) The rationale behind the variability in individual grid firing here bears some similarity to Dunn et al. (Dunn et al., 2017). This relationship needs to be clarified.

9) Quantification of the model predictions: Regarding the variability of individual grid firing fields, is it possible to quantify the effect and compare it to the publicly available grid cell dataset? Similarly, is it possible to quantify the model prediction on artificial mapping and put it into a format which would be more directly comparable to the data at the neural population level?

---

## [Author Response]

Essential revisions:1) The operation of the network should be presented more thoroughly. The development of this combined network is a major result, but it is a bit unclear how the network produces the reported results. In particular:a) Illustrating bump dynamics. The only dynamical data presented in the paper relates to path-integration and drift.

We added videos and supplementary figures with several panels in order to illustrate the dynamics of bump formation under different conditions.

First, for each of the three conditions presented in Figure 3, we now include a video showing one representative example (Figure 3—video 1, Figure 3—video 2, Figure 3—video 3, referenced in subsection “Coordination of the grid cell representation: suppression of incompatible drifts”), showing the dynamics of activity in all sub-networks over one second, starting from initialization. We have also added Figure 7—video 1 (referenced in the Discussion section), which shows the dynamics under grid cell depolarization, corresponding to the exact example shown in Figure 7—figure supplement 1A-D.

We added Figure 3—figure supplement 2 which includes quantitative analysis of bump score dynamics corresponding to each of the conditions from Figure 3 (referenced in subsection “Coordination of the grid cell representation: suppression of incompatible drifts”). We also added Figure 4—figure supplement 1 which includes the bump scores dynamics corresponding to the path-integration example from Figure 4A (referenced in subsection “Coordination of the grid cell representation: suppression of incompatible drifts”). Finally, we also added Figure 7—figure supplement 2, which includes a quantitative analysis of bump scores dynamics corresponding to each of the conditions from Figure 7 (referenced in the Discussion section).

When a bump develops from random initial conditions, does it do so steadily or in spurts?

Typically, as seen in the video for this condition, it is possible to identify very early on (shorter than the synaptic time scale, t) a single location within a single environment, in which a somewhat noisy bump state develops in the place cell network. This is followed by more gradual convergence to a persistent state as presented in Figure 2.

Similarly, how does bump score across different maps evolve when the bump develops into a mixed state?

This is now shown in Figure 7—figure supplement 2.

b) How many maps can be stored by the network? In the Supplementary Material, the authors calculate that this number scales with the number of neurons. Do simulations support this result, and what is the scaling prefactor?

Following this question, we extended our analysis to more thoroughly examine how quenched noise affects the number of maps that can be stored in the network. We have extensively revised the Appendix and added four Figures (Appendix—figure 1, Appendix—figure 2, Appendix—figure 3, Appendix— figure 4) related to the new text.

In our original submission, we analyzed how the quenched noise associated with place cell inputs to place cells scales with *N* (the number of place cells) and found that the variance is inversely proportional to *N*. This is now verified numerically in Appendix—figure 2A which also shows the scaling prefactor.

However, the variance of the quenched noise conveyed by grid cells to place cells does not similarly decrease with *n*, the number of grid cells. The difference in the two scaling behaviors is due to the difference in the type of remapping that occurs in place cells and grid cells: while the spatial relationships between place cells are chosen randomly in each map, grid cells within a module maintain fixed relative phases. This is verified numerically in Appendix—figure 2B (see also the expanded Appendix text and the Appendix—figure 1 and Appendix—figure 3 for covariance terms and variance in inputs conveyed to grid cells).

The outcome of this conclusion is that when neurons are added to the grid and place cell networks, the variance of quenched noise becomes dominated by the grid cell inputs, and saturates as the network sizes increase, thereby limiting the capacity (the number of spatial maps that can be stored in the network). Appendix—figure 4 provides an assessment of the capacity and shows qualitatively how it depends on the coupling parameter from grid cells to place cells. We qualitatively validated that even with the smallest coupling parameter chosen (𝛾_*g*_ =1), mixed states emerge under grid cell depolarization and path integration is performed successfully.

The conclusion that the transformation from grid cells to place cells limits the capacity of the network when place cell firing fields are assigned randomly, resonates well with recent results of Yim et al., (December 2019), which we now cite.

We added a paragraph in the Discussion section of the main manuscript that discusses the limitation on capacity, and how it relates to our assumptions on place cell and grid cell remapping.

c) What range of parameters, especially the relative strength of connections within and across subnetworks, produce desired results?

The relative strength of connections between grid cells and place cells (and vice versa) must indeed be set within a certain range in order to obtain the desired results: for example, if these connections are too weak, the network exhibits persistent states in which the bump positions in different sub-networks are incompatible. Moreover, it is essential for path integration and the coupling of grid cell modules, that both place cells can influence the position of grid cell bumps, and that grid cells can influence the state of the place cell network. Thus, it is not sufficient to include connections only in one direction. On the other hand, the connections cannot be too strong since we are interested in the regime in which the structure of steady states in each sub-network is predominantly determined by its internal connectivity.

In our numerical explorations we observed all these features qualitatively and settled on a set of parameters in which the desired properties were observed. This did not require fine tuning of the parameters. A quantitative demonstration of these points would require a very large number of simulations, since we would need to thoroughly explore the set of persistent states over a large range of parameters. We feel that such systematic exploration is interesting, but lies outside the main goals of our work, which are to demonstrate the qualitative consequences arising from bidirectional coupling between grid cells and place cell systems.

We added a discussion of these points (Subsection “coupling parameters”). Note also that Appendix—figure 4 includes an analysis of how 𝛾*_g_* affects the capacity.

2) How does the place cell network operate without grid cell input, and how does this compare to MEC lesion experiments? What activity patterns appear?

The hippocampal network in our model is set up such that it expresses bump activity patterns at the neural population level even without inputs from the MEC (see the Discussion section). This is in agreement with the observation that place fields are expressed under MEC lesions (Brun et al., 2008, Hales et al., 2014). Most likely, consistent assignment of the population activity patterns to spatial locations under MEC lesions relies on sensory inputs that project to the hippocampus from the LEC and are not modeled explicitly in our work. We added a paragraph on this topic in the Discussion section.

I presume that it would not path-integrate – is this true?

Correct, our model’s place cells activities won’t be updated in accordance with velocity inputs injected to grid cells in that case. This is consistent with the observation (Hales et al., 2014) that MEC lesions imparied spatial memory as the performance in the Morris water maze decreased under MEC lesion. Even without a lesion to the MEC, but with a disruption of grid cell activity by removing NMDA receptors, path-integration performance is impaired (Gil et al., 2018).

Also, it seems that place cell remapping arises from grid cell inputs, namely, map-dependent shifts in grid cell phase-is this true?

In our model, place cells are randomly assigned to locations in each environment, and random grid cell phases are assigned independently of the place cell remapping. Reciprocal connections between the two networks are then set up based on the overlap of desired joint activity patterns, as described in Methods section. Therefore, firing locations of place cells in a new environment are not determined by grid cell phases. Nevertheless, we expect that upon inducing persistent phase shifts in grid cells that correspond to a different embedded map, place cells will remap accordingly. We added a paragraph in the Discussion section to clarify this.

3) All the modeling presented in the paper is done in 1-d. It would be useful to confirm that it indeed also works in 2-d (and use the grid scales consistent with the experimental data). This would facilitate the comparison to (future) experimental results.

Conceptually, generalisation of the model to 2-d is straightforward. We implemented the model in 1-d for several reasons. First, the computational cost of simulations in 2-d is much higher than in 1-d. Second, visualization of the results (especially in multiple maps) is easier in 1-d than in 2-d, and since we do not expect fundamental differences between the two cases, in many ways it is advantageous for presentation purposes to work in 1-d.

We have recently been working on extending our results to 2-d, with the aim of adding additional realism into the model, and additional ingredients such as sensory inputs that project into the two networks. Qualitatively, we have been able to reproduce in 2-d the features demonstrated in the 1-d case: for example, we qualitatively verified the restriction of joint activity patterns to compatible persistent states, we were able to see path integration in the place cell network in response to velocity signal injected to grid cells, the coupling of grid cell modules in the presence of noise, and we observed, a similar outcome of grid cell depolarization on place cell activity patterns, as seen in Figure 7—video 1. However, extensive numerical work and analysis will be required for systematic quantification of all the results, and given the large volume of results already presented in the present manuscript we plan to include this analysis in a future study that will explore additional features of the model.

4) In addition to artificial remapping, where the firing rate distribution across grid fields is experimentally manipulated, a reorganization of the firing rate among grid fields can also be naturally observed in rate remapping paradigms (Diehl et al., 2017). Although the authors may have omitted this experimental finding because it does not fit into their framework of the place cell network mimicking global remapping, some of their analysis of artificial remapping shows that their network can in principle generate persistent mixed states. More in depth analysis should be added which describes to what extent the findings by Diehl et al. can be reproduced by the current network architecture or whether major modifications would be necessary. The Introduction, Results and Discussion sections should be expanded to include these analyses and their interpretation.

Thanks for pointing out that reorganization of the firing rate among grid fields occurs also under place cell rate remapping. This result is expected in our proposed model: changes in the firing rates of place cells, which occur under rate remapping, are expected to affect grid cell firing rates in a field-dependent manner due to the inhomogeneous connections from place cells to grid cells. While the phenomenon is similar to the results observed by Kanter et al., the origins for the changes in the firing rates of place cells are different in the two cases.

In order to observe reorganization of the firing rate among grid fields following rate remapping in our model we would need to explicitly model sensory inputs projecting into the hippocampus. We did not consider sensory inputs in the present work. The exploration of the MEC-hippocampal interaction with sensory inputs deserves a thorough analysis, and it would make sense to do this alongside an adaptation of the model to 2d environments. We believe that these extensions of the model should be addressed in a separate manuscript.

Our work is concerned with the interaction between grid cells and place cells and their joint dynamics as an attractor network: thus, our focus is on the ability of the MEC and hippocampus to sustain persistent patterns of activity at the population level irrespective of external inputs. This is what has led us to consider in detail the experiments of Kanter et al., where sensory inputs remained fixed under the experimental manipulation yet substantial changes were observed in the firing patterns in the hippocampus. Since sensory inputs were kept fixed, the modifications must have arisen from mechanisms that are intrinsic to the MEC-hippocampal network.

We now mention in the Results section the influence of place cell rate remapping on grid cell field variability. In addition, we added a new paragraph to the Discussion section in which we explain more explicitly why we focus on the experiment of Kanter et al.

5) Correction of drifts: It is interesting that the network can correct the drift of the grid module. There has now been a literature on this issue (Hardcastle, Ganguli and Giocomo, 2015; Mulas, Waniek and Conradt, 2016; Pollock et al., 2018). In particular these studies have shown that error correction can happen between the boundary cell or landmark cells to grid cells. It is unclear if/how the current error-correction mechanism fundamentally differs from the previous studies. It would be useful to clarify these issues.

The activation of boundary and landmark cells is typically related to sensory cues. In addition, the research on phase resets associated with encounters with walls has focused on single modules. The literature which is more relevant to the point raised in our manuscript addresses the need for an internal mechanism that could couple the phases of different grid cell modules independently from sensory inputs. The reason why such a mechanism is required, is that incompatible drifts (see below) in different modules are highly detrimental from the neural coding point of view, much more so than coordinated drifts (see Fiete, Burak and Wellinder 2008; Sreenivasan and Fiete, 2012; Burak, 2014; and Mosheiff and Burak, 2019 – all cited in our manuscript).

Our model does not include sensory inputs, yet we show that the reciprocal coupling between place cells and grid cells implements a mechanism that couples the distinct grid cell modules (Figure 5). Thus, our suggested error-correction mechanism is different from those mentioned in the comment (Hardcastle, Ganguli and Giocomo, 2015; Mulas, Waniek and Conradt, 2016; Pollock et al., 2018) since (1) it involves an internal mechanism which doesn’t depend on sensory inputs and because (2) it deals with coordination of the grid cell modules irrespective of the absolute represented location.

We have revised this part in the Introduction, clarifying that the error-correction mechanism we present is independent from sensory inputs (Results section). We clarified the difference from mentioned studies above in the Results section. We have also explained in more detail the concept of incompatible drifts (see below).

Also, how is "incompatible drift" defined in subsection “Coordination of the grid cell representation: suppression of incompatible drifts”?

Grid phases of two modules in two-dimensional space span a four-dimensional space (a two-dimensional phase for each module), of which the correct representations of position on the plane span a two dimensional subspace. In other words, during continuous motion in 2d, correctly updated phases span a 2d subspace of the 4d space of possible configurations. Similarly, in our 1d analogue, the phases of two modules span a two-dimensional space, whereas motion is one dimensional. During continuous motion, the change in the phase of one module must be related to the change in the phase of the other module through the ratio of their grid spacings, in order for the two modules to coherently represent a position in the local vicinity of the animal. Incompatible drifts are modifications in the phases, driven by dynamic noise, that do not obey these constraints. These issues are extensively discussed in several manuscripts that we cite, (e.g. Sreenivasan and Fiete, 2012, e.g. Burak, 2014, Mosheiff and Burak, 2019).

We notice that the labels in Figure 5A were ‘coherent/incoherent’. We changed these to ‘compatible/incompatible’ in order to use consistent terminology. In addition, we expanded the paragraph that introduces this topic in the Results section, to explain the concept of incompatible drifts.

6) Robustness with respect to the selection of the grid modules: The authors used [38.4,48, 64]cm for the 3 grid modules. Are the predictions robust with respect to the selection of the modules? For example, the form of decoding error using ML estimate would depend on the how the scales of the module are organized. With a hierarchical grid code (Mathis et al., 2012; Wei et al., 2015), it is much less prone to non-local errors, unlike the residual number system studied by Fiete et al., (2008).

Our specific grid spacings were chosen for convenience, such that an integer number of periods fits in the arena, which allows us to use periodic boundary conditions. Note also that the ratios of grid spacings (1.25 between modules 2 and 1; 1.33 between modules 3 and 2) are similar to those observed empirically.

Ratios of sequential grid spacings observed empirically by Stensola et al. were close to 1.4, when averaging across animals. In this respect, the spacings were similar to the idealized geometric series postulated by Mathis et al., and by Wei et al. However, across animals there is a large scatter in the ratios and in a given animal, the spacings are far from forming a precise geometric series.

With respect to the range of positions that can be represented by the grid cell code, Vágó and Ujfalussi (2018) concluded that when the number of modules is larger than 2, most choices of grid spacing ratios lead to near-optimal capacity.

With respect to the sensitivity to non-local readout errors, arising from incompatible phase errors: with biologically realistic choices of the grid spacings, the grid cells code is highly sensitive to such phase errors, even within a relatively small arena and with grid spacings that form a geometric series. As an example, see our recent paper (Mosheiff and Burak, 2019), where it was explicitly shown, both in 1d and in 2d, that suppression of incompatible phase errors dramatically improves the robustness of the code to noise, and that this is due to elimination of non-local readout errors.

7) One highlight of the current work is that the model could account for the seemingly paradoxical finding in Kanter et al., (2017). However, as the authors pointed out that there is also the issue regarding the generality of the finding in Kanter et al., (2017). Is there a regime that the proposed model can also account for Miao et al., (2015)? The authors wrote: "while in Ref (Kanter et al., 2017) chemogenetic receptors were targeted almost exclusively to layer II stellate cells of the MEC in transgenic animals, a broader population has been likely affected using viral injection in Ref (Miao et al., 2015)." The argument may not address the problem, because both stellate cells and pyramidal cells in EC2 can be grid cells.

Two central points that we meant to convey are (1) the specificity of the manipulation applied by Kanter et al., compared to the experiment of Miao et al., and (2) their simultaneous characterization of grid cell responses:

1) The manipulation used by Kanter et al., targeted the full extent of the MEC, with very little effect on neighboring tissues. In addition, the targeting was specific to excitatory cells. This was possible in the Kanter et al. experiment using transgenic mice in which DREADDs were specifically expressed almost exclusively in stellate cells of MEC layer II. On the other hand, the Miao et al. experiment relied on an injection of adeno-associated virus (AAV), which broadly targets neurons of various types: not only stellate and pyramidal cells but most likely also interneurons. In addition, the infection relied on the diffusion of the virus within the brain, which generates variability across the brain regions and in the number and types of neurons affected in each animal. Therefore, the affected cell types could not be precisely determined.

2) In the Kanter et al., experiment, the impact of DREADD activation on activity in MEC was probed experimentally, whereas similar electrophysiological measurements in the MEC were not performed in the experiment of Miao et al. Therefore, changes in MEC firing patterns following the experimental manipulation were not characterized.

This is now clarified in the revised manuscript (Discussion section).

Due to these reasons, the experimental manipulation of Miao et al., was less specific than the one applied in the Kanter et al. experiment, and its effect was less well characterized. An attempt to model the different conditions in this experiment, compared to the experiment of Kanter et al., requires information about the nature of the manipulation which is unavailable. Therefore, in our view, an attempt to do so would be too speculative.

These points need to be more carefully discussed, as it is critical to the test of the proposed model.

We provided an extensive discussion on possible experimental tests of our model, and specifically of our proposed explanation for the results of Kanter et al. The question of whether mixed states appear under the experimental protocols used by Kanter et al., is separate from the question why their results differ from those of Miao et al. Our predictions should be tested on the same transgenic animals used by Kanter et al., using similar experimental protocols.

8) The rationale behind the variability in individual grid firing here bears some similarity to Dunn et al. (Dunn et al., 2017). This relationship needs to be clarified.

Thanks for pointing out that the arXiv manuscript by Dunn et al., discussed a possible role of inputs from place cells in generating variability in individual grid cell firing fields.

The similarity between our mechanism and the one suggested by Dunn et al., is that they both arise from place cell inputs to grid cells. However, the exact sources which induce the variability are different: In Dunn et al., the source of variability stems from a random component added to the synaptic connectivity from place-like cells to grid cells. This would be similar to our model consisting only of a single map, but with (artificial) insertion of random noise in the connections between place cells to grid cells. Instead, we demonstrate a different principle: that simply by embedding and representing multiple spatial maps, variability in the firing fields of individual grid cells naturally and inevitably emerges due the quenched noise, and even without explicitly adding random noise from the type described by Dunn et al.

In our model, there are two contributions that elicit this variability:

1) Even with a non-stochastic and ideal connectivity from place cells to grid cells, embedding of multiple spatial maps inevitably embeds also quenched noise in the connectivity from place cells to grid cells. This is fundamentally a different source to the noise in the connections between place cells to grid cells than that of Dunn et al., although in practice, from a functional point of view it can have an equivalent effect.

2) In addition, the existence of the quenched noise and the recurrent place cell dynamics generate a spatially dependent distorted place cell bump and not an identical and smooth bump at each spatial location. Therefore, the variability of the place cell firing rates across distinct grid cell periodic locations inevitably contributes to the variability of individual grid cells across their firing fields. This distinction is emphasized in Figure 6A in our manuscript where place cell bumps which are slightly different from one another are shown at periodic grid spacing locations. Thus, since the place cell population activity pattern is different along the attractor, grid cell variability is inevitably exhibited by individual grid cells across their periodic firing fields.

We do not claim that our suggested mechanism is the sole determinant of individual grid cell variability. There could be multiple sources for variability in the grid fields (such as sensory inputs or connections between different modules), and one of them could also be random noise in the connections between place cells to grid cells as suggested by Dunn et al.

This is explained in the revised manuscript in the Discussion section. We added a reference to Dunn et al., in this context, alongside the work of Ismakov et al., and clarified the points raised above. We clarified our suggested mechanism which induces variability in individual grid cell firing fields in the Discussion section.

9) Quantification of the model predictions: Regarding the variability of individual grid firing fields, is it possible to quantify the effect and compare it to the publicly available grid cell dataset?

We did show a quantification in Figure 6B, which shows a histogram of the coefficient of variation (CV). This is similar to the quantification performed by Isamkov et al., presented in Figure 1B of their manuscript.

We showed that with the embedding of multiple spatial maps in the connectivity the CV increases. However, the purpose of our result is to point out a principle, and a specific and inevitable mechanism as a source, out of multiple possible sources, for the variability of individual grid cell firing fields.

We did not argue that this is the sole mechanism which is responsible for the experimentally observed variability and therefore a quantified comparison with the experimental results of Isamkov et al. is not straightforward. The experimental variability measured in Isamkov et al. reflects multiple sources which collectively account for the finding. This is now explained more explicitly in the Discussion section. We also discuss a way to experimentally eliminate the possible contribution of sensory inputs to the inter-field variability, and isolate variability that arises from intrinsic network mechanisms (Methods section).

It would be interesting to try and experimentally isolate the contribution to the grid firing field variability which arises only from the embedding of multiple spatial maps in order to make a clear comparison with theoretical prediction. We now briefly mention that this could perhaps be done by comparing animals that were exposed to a different number of spatial environments (Methods section).

Similarly, is it possible to quantify the model prediction on artificial mapping and put it into a format which would be more directly comparable to the data at the neural population level?

We did not fine-tune the model parameters to replicate exactly the statistics of responses reported by Kanter et al. It is probable that different statistics of the place cell responses would be obtained when varying the model parameters. Therefore, a direct comparison with the experimental data and the model’s artificial remapping statistics is not straightforward. At the population level, the responses of all place cells could be seen in Figure 8—figure supplement 1a. We now added a classification (explained in Methods section) of remapping types as described in the Results section.